# Stability analysis of heterogeneous oligopoly games of increasing players: A computational approach

**Ruirui Hou[1], Xiaoliang Li[2,3], Wenshuang Wan[1]\***

1 School of Management, Guangdong University of Science and Technology, Dongguan, China, 2 School of Business, Guangzhou College of Technology and Business, Guangzhou, China, 3 MoE Key Laboratory of Interdisciplinary Research of Computation and Economics, Shanghai University of Finance and Economics, Shanghai, China

\* xiaoliangbuaa@gmail.com

## Abstract

In this paper, we study an oligopolistic market endowed with an isoelastic demand function and a quadratic cost function, where heterogeneous firms coexist and produce the same product. We create new games by adding additional heterogeneous firms one after the other, and we examine the relative size of the stability region as the number of heterogeneous firms increases. For each model studied, we use the cylindrical algebraic decomposition method to analytically investigate the conditions for the local stability of the Cournot–Nash equilibrium. We find that the stability regions become larger as the number of heterogeneous firms involved increases. We also perform numerical simulations to investigate complex dynamics, such as periodic orbits and chaos, when the equilibrium loses its stability. Furthermore, we investigate the case of distinct cost parameters through numerical simulations and find that the dynamics seem more complicated than the case of identical cost parameters.

## 1 Introduction

In economic theory, markets are typically categorized into two polar structures: perfect competition and monopoly. In the former, numerous small firms compete against each other, whereas in the latter, a single producer controls the entire market. Between these extremes lies the oligopoly, characterized by a limited number of firms offering homogeneous products. The formal study of oligopoly dates back to Cournot [1], who introduced a static duopoly framework in which each firm is assumed to possess perfect knowledge of its rival's strategic behavior.

Several decades after Cournot's seminal work, Bertrand [2] introduced an alternative framework for modeling oligopolistic competition, in which firms compete by setting prices rather than quantities. More recently, Zhang et al. [3] investigated Cournot–Bertrand duopoly games under the assumption of bounded rationality,

**Data availability statement:** All relevant data are within the manuscript.

**Funding:** The second author is partially supported by Philosophy and Social Science Foundation of Guangdong (Grant No. GD25CLJ03), MoE Key Laboratory of Interdisciplinary Research of Computation and Economics (Shanghai University of Finance and Economics), and Innovation Team Project of Guangdong Colleges and Universities (Grant No. 2024WCXTD019). The funders had no role in study design, data collection and analysis, decision to publish, or preparation of the manuscript.

**Competing interests:** The authors have declared that no competing interests exist.

where players are rational agents who may strategically delay their decisions. Furthermore, when firms in the market are categorized as leaders and followers, the interaction is better described by a Stackelberg game. For more on Stackelberg competition, see, for example, [4–6].

Since Cournot's groundbreaking work, numerous studies have extended the Cournot model, particularly with a focus on its dynamic features. Theocharis [7] proposed a discrete dynamic game model based on a linear demand function, assuming that all firms are aware of the demand structure and operate under linear cost functions. He demonstrated that the market equilibrium remains stable only if the number of firms does not exceed three; beyond that, the system loses stability. Building on Theocharis' work, Fisher [8], McManus and Quandt [9] retained the linear demand function but relaxed the assumption of constant marginal costs. The key difference between their models lies in the adjustment mechanisms: McManus and Quandt [9] maintained Theocharis' best response approach, while Fisher [8] introduced bounded rationality, adopting an adaptive adjustment mechanism in response to market changes. A key finding of Fisher [8] as well as McManus and Quandt [9] is that the stability of the discrete dynamic model diminishes as the number of firms increases.

The dynamics of oligopoly games with an isoelastic demand function have been extensively studied in the existing literature. For instance, Ahmed and Agiza [10] extended the work of Puu [11] to encompass n competitors. In another contribution, Bischi et al. [12] proposed the local monopolistic approximation (LMA) adjustment mechanism, which assumes that firms make linear approximations of the market demand function. They showed that in models with isoelastic demand and constant marginal cost, the aggregate market output follows a logistic map, and that the system becomes unstable when the number of firms exceeds five. Building on this, Zhang and Gao [13] analyzed the effect of the LMA mechanism on the local stability of the Cournot equilibrium. Their results indicate that LMA enhances local stability when the inverse demand function is concave and reduces it when the function is convex. Additionally, Puu [14] introduced capacity constraints for firms and demonstrated that such constraints can prevent the destabilization typically caused by an increasing number of competitors.

The aforementioned studies primarily examine models in which competing firms adopt homogeneous decision-making mechanisms. However, in reality, it is uncommon for all firms to follow identical behavioral rules. A more realistic approach involves heterogeneous oligopoly models, where firms are assumed to use different strategies for decision-making. This perspective acknowledges that firms with varying business strategies—shaped by differences in risk preferences, information asymmetries, and operational goals—often coexist within the same industry. In contrast, industries characterized by more homogeneous firms, such as certain segments of the Internet sector, may experience only temporary coexistence before reaching equilibrium. As a result, there is a growing body of research focused on heterogeneous oligopoly models. See, for example, [15,16].

A key question in the study of heterogeneous oligopolies is whether increasing the number of firms leads to market destabilization. Tramontana et al. [17] addressed

this question and found a surprising result: the answer is negative. By progressively adding heterogeneous firms, they examined duopoly, triopoly, and quadropoly market structures. Unlike in markets composed of homogeneous firms—where stability tends to deteriorate as more firms enter—their findings suggest that increasing the number of heterogeneous firms can actually enhance the stability of the market equilibrium. This outcome supports the notion that in competitive environments, firms employing diverse strategies are more likely to coexist and contribute to market stability, whereas firms with similar strategies are more prone to elimination and susceptible to market fluctuations.

Our research yields similar results to those of Tramontana et al. [17] but with a more evident conclusion: as additional heterogeneous firms enter the market, the region for the local stability of the equilibrium expands. In contrast to the mathematical approach of [17], our study employs the cylindrical algebraic decomposition (CAD) method, a symbolic computational tool that can be used to select sample points of semi-algebraic sets. It should be emphasized that this computational tool is symbolic in the sense that its results are exact and error-free, making it well-suited for discovering and proving economic theorems, as discussed in [18–20]. In comparison, numerical methods are widely used in the computation of equilibria. However, numerical methods have several shortcomings: first, numerical computation may encounter the problem of instability, which could make the results completely useless; second, most numerical algorithms only search for a single equilibrium and are nearly infeasible for detecting multiple equilibria.

We employ the CAD method to systematically and automatically determine the local stability conditions for the equilibria of the models examined in this paper, providing clearer proofs than traditional pencil-and-paper approaches. Moreover, the 5-firm oligopoly game analyzed in our study features perfectly rational players—an aspect not considered by Tramontana et al. [17]. We conduct numerical simulations to explore the complex dynamics that emerge when the game loses stability, including periodic orbits and chaotic behaviors. In addition, we investigate scenarios with heterogeneous cost parameters and find that the resulting dynamics are even more intricate than those observed with identical cost parameters.

The remainder of this paper is organized as follows. Sect 2 examines a duopoly game consisting of a gradient-adjustment player and a naive player. In Sect 3, an adaptive player is introduced into the oligopolistic competition. Sect 4 extends the analysis to a quadropoly by incorporating an LMA player. In Sect 5, a fully rational player is finally considered. Sect 6 discusses the contributions of this paper and the economic implications of our findings. Finally, Sect 7 provides concluding remarks.

## 2 Game of two firms

Motivated by Tramontana et al. [17], we investigate markets where firms adopt heterogeneous decision-making mechanisms and produce the same product. Let $q_i(t)$ denote the output of firm $i$ at period $t$. We assume that each firm's cost function is quadratic, i.e.,

$$C_i(q_i) = cq_i^2, \tag{1}$$

where $c > 0$ is a uniform parameter for all firms. Here, we employ the nonlinear cost function rather than the linear cost function of Tramontana et al. [17] since the relevant results under diseconomies of scale are focused on in our study. In addition, we assume that the market is featured by an isoelastic demand function introduced by Puu [11], which is based on the hypothesis of the Cobb–Douglas utility function by the agents. Specifically, the market inverse demand function is

$$p(Q) = \frac{1}{Q}, \tag{2}$$

where $Q = \sum_i q_i$ represents the total supply of the market. Simple calculations show that the price elasticity of the market demand is exactly one. In addition, interested readers can see Appendix of [21] for the microeconomic foundations of the general isoelastic demand function.

First, consider a duopoly game in which the first firm adopts a *gradient adjustment mechanism*, while the second firm follows a *naive expectation mechanism*. Both mechanisms reflect bounded rationality. Specifically, the first firm adjusts its output based on the marginal profit observed in the previous period $t$, increasing or decreasing its output accordingly. It then determines its output for period $t + 1$ based on this information as follows:

$$q_1(t + 1) = q_1(t) + kq_1(t)\frac{\partial \Pi_1(t)}{\partial q_1(t)}, \tag{3}$$

where $\Pi_1(t) = \frac{q_1(t)}{q_1(t) + q_2(t)} - cq_1^2(t)$ is the profit of firm 1 as period $t$, and $k > 0$ is a parameter controlling the adjustment speed.

The second firm knows exactly the form of the price function and can therefore estimate its profit at period $t + 1$ to be

$$\Pi_2^e(t + 1) = \frac{q_2(t + 1)}{q_1^e(t + 1) + q_2(t + 1)} - cq_2^2(t + 1), \tag{4}$$

where $q_1^e(t + 1)$ is its expectation of firm 1's output at period $t + 1$. It is reasonable to assume that firm 2 has no knowledge of its rival's production plan for the current period. We therefore assume that firm 2 is a naive player, expecting its competitor to produce the same quantity as in the previous period, i.e., $q_1^e(t + 1) = q_1(t)$. Hence,

$$\Pi_2^e(t + 1) = \frac{q_2(t + 1)}{q_1(t) + q_2(t + 1)} - cq_2^2(t + 1). \tag{5}$$

To maximize the expected profit, the second firm attempts to solve the first condition $\partial \Pi_2^e(t + 1)/\partial q_2(t + 1) = 0$, i.e.,

$$q_1(t) - 2\, cq_2(t + 1)(q_1(t) + q_2(t + 1))^2 = 0. \tag{6}$$

It is worth noting that Eq (6) is a cubic polynomial. While a general cubic polynomial can have up to three real roots, it is straightforward to verify that (6) admits a unique real solution for $q_2(t + 1)$. However, the closed-form expression $q_2(t + 1) = \Phi(q_1(t))$ is particularly intricate, where

$$\Phi(x) = \frac{2^{\frac{1}{3}}\left(c^2x\left(4\,cx^2 + 3\sqrt{3}\sqrt{8\,cx^2 + 27} + 27\right)\right)^{\frac{1}{3}}}{6\,c} - \frac{2\,x}{3}$$
$$+ \frac{2^{\frac{2}{3}}cx^2}{3\left(c^2x\left(4\,cx^2 + 3\sqrt{3}\sqrt{8\,cx^2 + 27} + 27\right)\right)^{\frac{1}{3}}}. \tag{7}$$

However, we assume that firm 2, by observing its rival's output in the previous period, possesses the computational ability to determine its best response, denoted by $R_2(q_1(t))$. Clearly, $R_2(q_1(t)) = \Phi(q_1(t))$. As a result, the model can be formulated as the following discrete dynamical system.

$$T_{GN}(q_1, q_2) : \begin{cases} q_1(t + 1) = q_1(t) + kq_1(t)\left[\dfrac{q_2(t)}{(q_1(t) + q_2(t))^2} - 2\,cq_1(t)\right], \\ q_2(t + 1) = R_2(q_1(t)). \end{cases} \tag{8}$$

By setting $q_1(t+1) = q_1(t) = q_1^*$ and $q_2(t+1) = q_2(t) = q_2^*$, the equilibrium can be identified by

$$\begin{cases} q_1^* = q_1^* + kq_1^*\left(\dfrac{q_2^*}{(q_1^* + q_2^*)^2} - 2\,cq_1^*\right), \\ q_2^* = R_2(q_1^*), \end{cases} \tag{9}$$

where $q_2^* = R_2(q_1^*)$ can be reformulated into $q_1^* - 2\,cq_2^*(q_1^* + q_2^*)^2 = 0$ according to (6). Thus, we have

$$\begin{cases} kq_1^*\left(\dfrac{q_2^*}{(q_1^* + q_2^*)^2} - 2\,cq_1^*\right) = 0, \\ q_1^* - 2\,cq_2^*(q_1^* + q_2^*)^2 = 0, \end{cases} \tag{10}$$

which can be solved by a unique solution (the Cournot–Nash equilibrium)

$$E_{GN}^1 = \left(\frac{1}{\sqrt{8c}}, \frac{1}{\sqrt{8c}}\right). \tag{11}$$

We note that $(0,0)$ is not an equilibrium since it is not defined for the iteration map (8). In order to investigate the local stability of an equilibrium $(q_1^*, q_2^*)$, we consider the Jacobian matrix

$$\boldsymbol{J}_{GN}(q_1^*, q_2^*) = \begin{bmatrix} \dfrac{\partial q_1(t+1)}{\partial q_1(t)}\Big|_{(q_1^*,q_2^*)} & \dfrac{\partial q_1(t+1)}{\partial q_2(t)}\Big|_{(q_1^*,q_2^*)} \\ \dfrac{\partial q_2(t+1)}{\partial q_1(t)}\Big|_{(q_1^*,q_2^*)} & \dfrac{\partial q_2(t+1)}{\partial q_2(t)}\Big|_{(q_1^*,q_2^*)} \end{bmatrix}, \tag{12}$$

where

$$\begin{aligned} \frac{\partial q_1(t+1)}{\partial q_1(t)}\Big|_{(q_1^*,q_2^*)} &= 1 + kq_2^*\frac{q_2^* - q_1^*}{(q_1^* + q_2^*)^3} - 4\,ckq_1^*, \\ \frac{\partial q_1(t+1)}{\partial q_2(t)}\Big|_{(q_1^*,q_2^*)} &= kq_1^*\frac{q_1^* - q_2^*}{(q_1^* + q_2^*)^3}. \end{aligned} \tag{13}$$

Furthermore, the derivative of $q_2(t+1)$ with respect to $q_2(t)$ is 0 as $R_2$ does not involve $q_2$. However, the derivative of $q_2(t+1)$ with respect to $q_1(t)$ may not be directly obtained. We know that $q_2(t+1) = R_2(q_1(t))$ in (8) is equivalent to (6). Thus, according to the implicit differentiation, from (6) we have

$$\begin{aligned} & 1 - 2\,c(q_1(t) + q_2(t+1))^2\frac{\partial q_2(t+1)}{\partial q_1(t)} \\ & - 4\,cq_2(t+1)(q_1(t) + q_2(t+1))\left(1 + \frac{\partial q_2(t+1)}{\partial q_1(t)}\right) = 0, \end{aligned} \tag{14}$$

which implies that

$$\frac{\partial q_2(t+1)}{\partial q_1(t)}\Big|_{(q_1^*,q_2^*)} = -\frac{4\,cq_1^*q_2^* + 4\,cq_2^{*2} - 1}{2\,c(q_1^{*2} + 4\,q_1^*q_2^* + 3\,q_2^{*2})}. \tag{15}$$

Therefore, it can be acquired that

$$J_{GN}(q_1^*, q_2^*) = \begin{bmatrix} 1 + kq_2^* \frac{q_2^* - q_1^*}{(q_1^* + q_2^*)^3} - 4\,ckq_1^* & kq_1^* \frac{q_1^* - q_2^*}{(q_1^* + q_2^*)^3} \\ -\frac{4\,cq_1^* q_2^* + 4\,cq_2^{*2} - 1}{2\,c(q_1^{*2} + 4\,q_1^* q_2^* + 3\,q_2^{*2})} & 0 \end{bmatrix},$$

(16)

At $E_{GN}^1 = (1/\sqrt{8c}, 1/\sqrt{8c})$, it is derived that

$$J_{GN}(E_{GN}^1) = \begin{bmatrix} 1 - k\sqrt{2\,c} & 0 \\ 0 & 0 \end{bmatrix}.$$

(17)

Obviously, its eigenvalues are $\lambda_1 = 1 - k\sqrt{2\,c}$ and $\lambda_2 = 0$. Hence, $E_{GN}^1$ is locally stable if $k\sqrt{c} < \sqrt{2}$. We formally state the above results in the following theorem.

**Theorem 1.** *Model $T_{GN}$ described by map* (8) *has a unique equilibrium*

$$\left( \frac{1}{\sqrt{8c}}, \frac{1}{\sqrt{8c}} \right),$$

(18)

*which is locally stable if*

$$k\sqrt{c} < \sqrt{2}.$$

(19)

According to the theorem above, the model becomes unstable when the value of parameter $k$ or $c$ is sufficiently large. In other words, the slower firm 1 adjusts its output, or the smaller the cost parameter $c$ is, the more stable the game's equilibrium becomes. This result is consistent with findings from studies on Cournot duopoly games with linear cost functions. See, for example, [21–23].

Fig 1(a) depicts the one-dimensional bifurcation diagram of model $T_{GN}$ to illustrate the dynamic behaviors in the destabilization of the equilibrium if we vary the value of the adjustment speed $k$. In numerical simulations of Fig 1(a), we fix $c = 1.0$ and choose the initial iteration state to be $(q_1(0), q_2(0)) = (0.1, 0.1)$. The diagram against $q_1$ and $q_2$ is marked in red and blue, respectively. It can be observed that the equilibrium loses its stability through a flip bifurcation. That is, the equilibrium transitions to chaotic dynamics through a cascade of period-doubling bifurcations [24,25]. Here, we should emphasize that the adjustment speed $k$ of firm 1 is a destabilizing factor of the game, which has already been evidenced in [8]. Furthermore, we find that the amplitude of periodic or chaotic orbits of firm 1's output, i.e., $q_1$, is much larger than that of firm 2's output, i.e., $q_2$. From an economic point of view, this means that when the market is in a state of disarray, firms adopting the gradient adjustment mechanism may show more violent reactions than those adopting the naive expectation. In addition, Fig 1(b) reports the one-dimensional bifurcation diagram with respect to $c$, where we fix $k = 1.0$ and set $(q_1(0), q_2(0)) = (0.1, 0.1)$. Similarly, we also observe the route to chaotic dynamics through a cascade of period-doubling bifurcations. One can see that a high level of the cost parameter $c$ of the involved firms leads to the instability of the game.

In Fig 1(c), we provide the two-dimensional bifurcation diagram of model $T_{GN}$, where we consider the effects of the two parameters, namely the cost parameter of the two firms, i.e., $c$, and the adjustment speed of firm 1, i.e., $k$. Readers can refer to [26] for more information regarding the two-dimensional bifurcation diagram. In numerical simulations of producing these bifurcation diagrams, we choose the initial state to be $(q_1(0), q_2(0)) = (0.1, 0.1)$. Parameter points corresponding to periodic orbits with different orders are marked in different colors and are marked in yellow if the order is greater than or equal to 24. In addition, we use the grey color to mark parameter points corresponding to the trajectory diverges

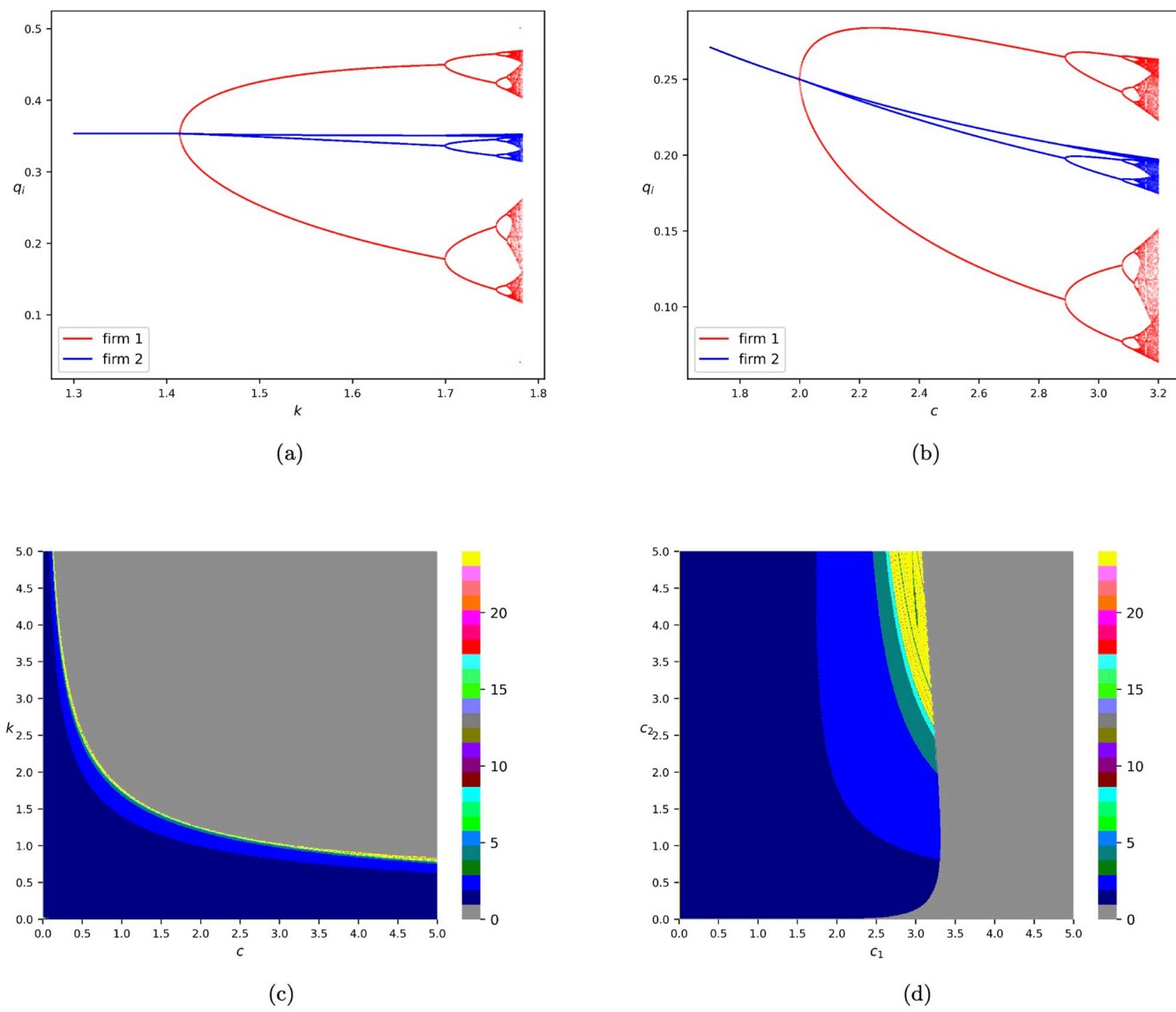

**Fig 1. Bifurcation diagrams of model** $T_{GN}$. (a) One-dimensional bifurcation diagram with respect to $k$ by fixing $c = 1.0$. (b) One-dimensional bifurcation diagram with respect to $c$ by fixing $k = 1.0$. The diagram against $q_1, q_2$ is marked in red and blue, respectively. (c) Two-dimensional bifurcation diagram with respect to $c$ and $k$. (d) Two-dimensional bifurcation diagram with respect to $c_1$ and $c_2$ if we fix $k = 1.0$ and assume that the cost parameters of the two firms are $c_1$ and $c_2$, respectively. All the numerical simulations are conducted by choosing the initial iteration state to be $(q_1(0), q_2(0)) = (0.1, 0.1)$.

(approaches $\infty$). Accordingly, the yellow points may be viewed as the parameter values where complex dynamics such as chaos or periodic solutions with high orders take place. It can be observed that the equilibrium loses its stability through a series of period-doubling bifurcations as the value of $c$ or $k$ increases.

Furthermore, from a theoretical perspective, it is interesting to investigate the dynamic transitions of the model if the cost parameters of firms are distinct. Accordingly, we conduct numerical simulations by fixing $k = 1.0$ and assume that the cost parameters of the two firms are $c_1$ and $c_2$, respectively. In Fig 1(d), we report the two-dimensional bifurcation diagram based on the results of our numerical simulations. We find that the ratio of $c_1$ to $c_2$ plays an ambitious role in affecting the dynamic behaviors of the game. If the value of $c_1$ is relatively small compared to that of $c_2$, then there exist complex

dynamics such as chaos in the destabilization of the equilibrium. However, if the value of $c_1$ is sufficiently large compared to that of $c_2$, the trajectories may transition from converging to the equilibrium directly to diverging to $\infty$. In this case, no periodic or chaotic orbits even appear.

## 3 Game of three firms

In this section, we introduce a new player with bounded rationality and incorporate it into the model from the previous section. This player adopts an *adaptive mechanism*, meaning that at each period $t+1$, it determines its output $q_3(t+1)$ based on its previous output $q_3(t)$ and its best response to the expected actions of the other two competitors. It is assumed that this player naively expects firms 1 and 2 to produce the same quantities as at period $t$. Accordingly, the third firm calculates its best response $R_3(q_1(t), q_2(t))$ in order to maximize its expected profit. Specifically, $R_3(q_1(t), q_2(t))$ is defined as the solution for $q_3'(t+1)$ of the equation

$$q_1(t) + q_2(t) - 2\,cq_3'(t+1)(q_1(t) + q_2(t) + q_3'(t+1))^2 = 0. \tag{20}$$

The closed-form expression of $R_3(q_1(t), q_2(t))$ is quite complicated. One can verify that $R_3(q_1(t), q_2(t)) = \Phi(q_1(t) + q_2(t))$ (see Eq (7) for the details of $\Phi$). The adaptive decision mechanism for firm 3 is that it chooses the output $q_3(t+1)$ proportionally to be

$$q_3(t+1) = (1 - l)q_3(t) + lR_3(q_1(t), q_2(t)), \tag{21}$$

where $l \in (0, 1]$ is a parameter controlling the proportion. Similarly, we have the reaction function of firm 2 is $R_2(q_1(t), q_3(t)) = \Phi(q_1(t) + q_3(t))$.

Hence, the triopoly competition can be described by

$$T_{GNA}(q_1, q_2, q_3) :$$
$$\begin{cases} q_1(t+1) = q_1(t) + kq_1(t)\left[\dfrac{q_2(t) + q_3(t)}{(q_1(t) + q_2(t) + q_3(t))^2} - 2\,cq_1(t)\right], \\ q_2(t+1) = R_2(q_1(t), q_3(t)), \\ q_3(t+1) = (1 - l)q_3 + lR_3(q_1(t), q_2(t)). \end{cases} \tag{22}$$

One can see that an equilibrium $(q_1^*, q_2^*, q_3^*)$ of the dynamic game satisfies that

$$\begin{cases} kq_1^*\left(\dfrac{q_2^* + q_3^*}{(q_1^* + q_2^* + q_3^*)^2} - 2\,cq_1^*\right) = 0, \\ q_1^* + q_3^* - 2\,cq_2^*(q_1^* + q_2^* + q_3^*)^2 = 0, \\ q_1^* + q_2^* - 2\,cq_3^*(q_1^* + q_2^* + q_3^*)^2 = 0, \end{cases} \tag{23}$$

which can be solved by

$$E_{GNA}^1 = \left(0, \frac{1}{\sqrt{8c}}, \frac{1}{\sqrt{8c}}\right),$$
$$E_{GNA}^2 = \left(\frac{1}{\sqrt{9c}}, \frac{1}{\sqrt{9c}}, \frac{1}{\sqrt{9c}}\right). \tag{24}$$

At an equilibrium $(q_1^*, q_2^*, q_3^*)$, the Jacobian matrix of $T_{GNA}$ takes the form

$$J_{GNA}(q_1^*, q_2^*, q_3^*) = \begin{bmatrix} \frac{\partial q_1(t+1)}{\partial q_1(t)}\Big|_{(q_1^*,q_2^*,q_3^*)} & \frac{\partial q_1(t+1)}{\partial q_2(t)}\Big|_{(q_1^*,q_2^*,q_3^*)} & \frac{\partial q_1(t+1)}{\partial q_3(t)}\Big|_{(q_1^*,q_2^*,q_3^*)} \\ \frac{\partial q_2(t+1)}{\partial q_1(t)}\Big|_{(q_1^*,q_2^*,q_3^*)} & \frac{\partial q_2(t+1)}{\partial q_2(t)}\Big|_{(q_1^*,q_2^*,q_3^*)} & \frac{\partial q_2(t+1)}{\partial q_3(t)}\Big|_{(q_1^*,q_2^*,q_3^*)} \\ \frac{\partial q_3(t+1)}{\partial q_1(t)}\Big|_{(q_1^*,q_2^*,q_3^*)} & \frac{\partial q_3(t+1)}{\partial q_2(t)}\Big|_{(q_1^*,q_2^*,q_3^*)} & \frac{\partial q_3(t+1)}{\partial q_3(t)}\Big|_{(q_1^*,q_2^*,q_3^*)} \end{bmatrix}. \tag{25}$$

The first and the second rows of the matrix could be similarly computed as Sect 2. For the third row, we have

$$\begin{aligned} \frac{\partial q_3(t+1)}{\partial q_1(t)} &= l\frac{\partial R_3(q_1(t), q_2(t))}{\partial q_1(t)}, \\ \frac{\partial q_3(t+1)}{\partial q_2(t)} &= l\frac{\partial R_3(q_1(t), q_2(t))}{\partial q_2(t)}, \\ \frac{\partial q_3(t+1)}{\partial q_3(t)} &= 1 - l, \end{aligned} \tag{26}$$

where $\partial R_3(q_1(t), q_2(t))/\partial q_1(t)$ and $\partial R_3(q_1(t), q_2(t))/\partial q_2(t)$ can be acquired using the method of implicit differentiation. From an economic point of view, we ignore the boundary equilibrium $E_{GNA}^1$ but only consider the positive (Cournot–Nash) equilibrium $E_{GNA}^2$, where the Jacobian matrix becomes

$$J_{GNA}(E_{GNA}^2) = \begin{bmatrix} 1 - 10\,k\sqrt{c}/9 & -k\sqrt{c}/9 & -k\sqrt{c}/9 \\ -1/10 & 0 & -1/10 \\ -l/10 & -l/10 & 1-l \end{bmatrix}. \tag{27}$$

Let $A(\lambda)$ denote the characteristic polynomial of a Jacobian matrix $J$. The eigenvalues of $J$ are precisely the roots of $A(\lambda)$. Therefore, the stability analysis of the system reduces to determining whether all roots of $A$ lie within the open unit disk, i.e., $|\lambda| < 1$. To the best of our knowledge, beyond the Routh–Hurwitz criterion [27], which extends the classical criterion for continuous-time systems, two additional methods are commonly used for stability analysis in discrete dynamical systems: the Schur–Cohn criterion [28, pp. 246–248] and the Jury criterion [29]. In the following, we present a brief review of the Schur–Cohn criterion.

**Proposition 1** (Schur-Cohn Criterion). *Consider an n-dimensional discrete dynamical system, and suppose that the characteristic polynomial of its Jacobian matrix is given by*

$$A = \lambda^n + a_{n-1}\lambda^{n-1} + \cdots + a_0. \tag{28}$$

*Denote*

$$D_i^{\pm} = \left\| \begin{pmatrix} 1 & a_{n-1} & a_{n-2} & \cdots & a_{n-i+1} \\ 0 & 1 & a_{n-1} & \cdots & a_{n-i+2} \\ 0 & 0 & 1 & \cdots & a_{n-i+3} \\ \vdots & \vdots & \vdots & \ddots & \vdots \\ 0 & 0 & 0 & \cdots & 1 \end{pmatrix} \pm \begin{pmatrix} a_{i-1} & a_{i-2} & \cdots & a_1 & a_0 \\ a_{i-2} & a_{i-3} & \cdots & a_0 & 0 \\ \vdots & \vdots & \ddots & \vdots & \vdots \\ a_1 & a_0 & \cdots & 0 & 0 \\ a_0 & 0 & \cdots & 0 & 0 \end{pmatrix} \right\|. \tag{29}$$

*The characteristic polynomial A has all its roots inside the unit open disk if and only if the following two statements hold:*

1. $A(1) > 0$ *and* $(-1)^n A(-1) > 0$,
2. $D_1^\pm > 0, D_3^\pm > 0, \ldots, D_{n-3}^\pm > 0, D_{n-1}^\pm > 0$ *(when n is even), or*

   $D_2^\pm > 0, D_4^\pm > 0, \ldots, D_{n-3}^\pm > 0, D_{n-1}^\pm > 0$ *(when n is odd).*

**Corollary 1.** *Consider a 3-dimensional discrete dynamical system whose Jacobian matrix has a characteristic polynomial of the form*

$$A = \lambda^3 + a_2 \lambda^2 + a_1 \lambda + a_0. \tag{30}$$

*An equilibrium E is locally stable if the following inequalities are satisfied at E.*

$$\begin{cases} 1 + a_2 + a_1 + a_0 > 0, \\ 1 - a_2 + a_1 - a_0 > 0, \\ -a_0^2 - a_0 a_2 + a_1 + 1 > 0, \\ -a_0^2 + a_0 a_2 - a_1 + 1 > 0. \end{cases} \tag{31}$$

For the 3-dimensional discrete dynamic system (22), one can verify that at the Cournot–Nash equilibrium $E_{GNA}^2$, the characteristic polynomial is

$$A = \lambda^3 - \left(2 - l - \frac{10k\sqrt{c}}{9}\right)\lambda^2 - \left(-\frac{11lk\sqrt{c}}{10} + \frac{101l}{100} - 1 + \frac{101k\sqrt{c}}{90}\right)\lambda \\ - \frac{lk\sqrt{c}}{50} + \frac{l}{100} + \frac{k\sqrt{c}}{90}. \tag{32}$$

Accordingly, the local stability condition (31) can be reformulated into

$$CD_{GNA}^1 > 0, \ CD_{GNA}^2 > 0, \ CD_{GNA}^3 < 0, \ CD_{GNA}^4 < 0, \tag{33}$$

where

$$\begin{aligned} CD_{GNA}^1 &= kl\sqrt{c}, \\ CD_{GNA}^2 &= 504\,kl\sqrt{c} - 1010\,k\sqrt{c} - 909\,l + 1800, \\ CD_{GNA}^3 &= 324\,ck^2l^2 - 18360\,ck^2l + 10100\,ck^2 - 16524\,kl^2\sqrt{c} - 840420\,kl\sqrt{c} \\ &\quad + 8181\,l^2 + 891000\,k\sqrt{c} + 801900\,l - 1620000, \\ CD_{GNA}^4 &= 36\,ck^2l^2 + 1960\,ck^2l + 1764\,kl^2\sqrt{c} - 1100\,ck^2 + 93420\,kl\sqrt{c} \\ &\quad - 99000\,k\sqrt{c} - 891\,l^2 - 89100\,l. \end{aligned} \tag{34}$$

Firstly, we find that $CD_{GNA}^1 > 0$ can be ignored since it is always true for all feasible parameter values, namely $k > 0$, $c > 0$, and $1 \geq l > 0$. A further question is whether the other three inequalities could be simplified. To answer this question, we might investigate the inclusion relations of these inequalities. It worth noticing that the surfaces $CD_{GNA}^2 = 0$, $CD_{GNA}^3 = 0$, and $CD_{GNA}^4 = 0$ divide the parameter space $\{(k, l, c) \mid k > 0, 1 \geq l > 0, c > 0\}$ of our concern into a number of separated regions. Moreover, the signs of $CD_{GNA}^i$ ($i = 1, 2, 3, 4$) are invariant in a given region. This means that in each of these regions, we can identify whether the inequalities in (33) are satisfied by checking them at a single sample point. In simple cases, the selection of sample points might be done manually. Generally, however, the selection is extremely complicated and can be automated by using the CAD method.

Below, theorem proofs mostly rely on the CAD method, which is a symbolic computation method. The reader will see that the introduction of the CAD method allows the deduction of stability conditions of the considered models to be conducted systematically and automatically, thus greatly simplifying the theorem proving. The CAD method is the first practical quantifier elimination algorithm, proposed by Collins [30], and is therefore also known as Collins' algorithm. This algorithm decomposes any semi-algebraic set in the $n$-dimensional real number space $\mathbb{R}^n$ into a finite number of disjoint semi-algebraic sets. All the resulting semi-algebraic sets are defined by the same set of polynomials, and the sign of the polynomials defined on each semi-algebraic set remains unchanged. The original CAD method was not efficient enough and was later improved by Brown [31], Collins and Hong [32]. Specifically, the CAD method can compute the cylindrical algebraic decomposition and its sample point set on $\mathbb{R}^n$, such that the signs of the given polynomials are invariant in each decomposition. More details can be found in, e.g., [30–32].

In Table 1, we list all the selected sample points generated by the CAD method such that there exists at least one point in each region divided by the surfaces $CD^2_{GNA} = 0$, $CD^3_{GNA} = 0$, and $CD^4_{GNA} = 0$. The four inequalities in (33) are verified at these sample points one by one, the results of which are also reported in Table 1. It is found that at the sample points where $CD^2_{GNA} > 0$ is true, the other three inequalities are also true. Hence, if $CD^2_{GNA} > 0$ is satisfied, then all the four inequalities in (33) will be satisfied definitely. In other words, only $CD^2_{GNA} > 0$ is needed herein to determine the local stability of $E^2_{GNA}$. It is evident that $CD^2_{GNA} > 0$ is equivalent to

$$k\sqrt{c} < \frac{9(101\,l - 200)}{2(252\,l - 505)}. \tag{35}$$

Therefore, we summarize the obtained results in the following theorem.

**Theorem 2.** *Model $T_{GNA}$ described by map* (22) *has a unique positive equilibrium*

$$\left( \frac{1}{\sqrt{9c}}, \frac{1}{\sqrt{9c}}, \frac{1}{\sqrt{9c}} \right), \tag{36}$$

*which is locally stable if*

$$k\sqrt{c} < \frac{9(101\,l - 200)}{2(252\,l - 505)}. \tag{37}$$

The one-dimensional bifurcation diagram of model $T_{GNA}$ with respect to $k$ by fixing $c = 1.0$ and $l = 0.5$ is plotted in Fig 2(a), where numerical simulations are conducted by choosing the initial iteration state to be

**Table 1. Stability conditions of $T_{GNA}$ at selected sample points.**

| sample point of $(k, l, c)$ | $CD^1_{GNA} > 0$ | $CD^2_{GNA} > 0$ | $CD^3_{GNA} < 0$ | $CD^4_{GNA} < 0$ |
|---|---|---|---|---|
| (455/256, 71/256, 1/4) | true | true | true | true |
| (31/8, 71/256, 1/4) | true | false | true | true |
| (601/128, 71/256, 1/4) | true | false | false | true |
| (453/256, 183/256, 1/4) | true | true | true | true |
| (1439/256, 183/256, 1/4) | true | false | true | true |
| (1577/16, 183/256, 1/4) | true | false | false | true |
| (49855/256, 183/256, 1/4) | true | false | true | true |
| (25673/128, 183/256, 1/4) | true | false | true | false |
| (451/256, 15/16, 1/4) | true | true | true | true |
| (5237/256, 15/16, 1/4) | true | false | true | true |
| (2425/64, 15/16, 1/4) | true | false | true | false |

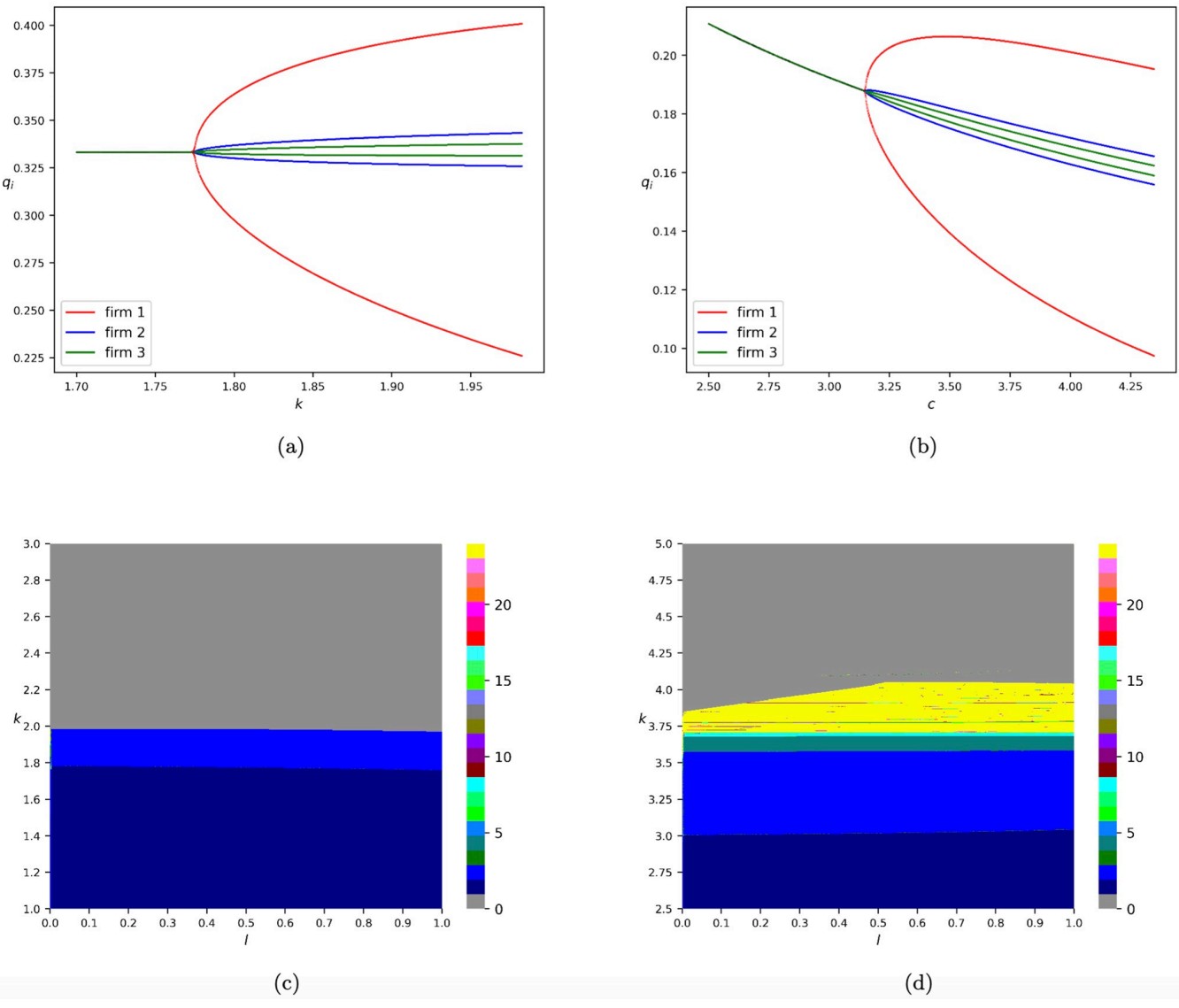

**Fig 2**. **Bifurcation diagrams of model** $T_{GNA}$. (a) One-dimensional bifurcation diagram with respect to $k$ by fixing $c = 1.0$ and $l = 0.5$. (b) One-dimensional bifurcation diagram with respect to $c$ by fixing $k = 1.0$ and $l = 0.5$. The diagram against $q_1, q_2, q_3$ is marked in red, blue, and green, respectively. (c) Two-dimensional bifurcation diagram with respect to $l$ and $k$ by fixing $c = 1.0$. (d) Two-dimensional bifurcation diagram with respect to $l$ and $k$ if we set the cost parameters of the three firms to be 0.2, 1.0, and 1.0, respectively. All the numerical simulations are conducted by choosing the initial iteration state to be $(q_1(0), q_2(0), q_3(0)) = (0.1, 0.1, 0.1)$.

$(q_1(0), q_2(0), q_3(0)) = (0.1, 0.1, 0.1)$. One can see that the equilibrium is stable when $k < 1.775879397$ and loses its stability through a flip bifurcation. In contrast to model $T_{GN}$, the most complex dynamics taking place in model $T_{GNA}$ are 2-cycle orbits, and there are not any periodic orbits with higher orders or chaos. When the value of $k$ increases to 1.986934673, 2-cycle orbits transition to trajectories that diverge. In addition, we also find that the output amplitude of the first player is much larger than the other two players in model $T_{GNA}$, which is similar to model $T_{GN}$. Fig 2(b) depicts the one-dimensional bifurcation diagram of model $T_{GNA}$ with respect to $c$ by fixing $k = 1.0$ and $l = 0.5$, where similar dynamics can be found.

The results regarding other values of the adaptive proportion parameter $l$ can be found in Fig 2(c). We find that both the adjustment speed $k$ of firm 1 and the cost parameter $c$ are destabilizing factors of the equilibrium.

Furthermore, Fig 2(d) illustrates the dynamic transitions of model $T_{GNA}$ when the firms have heterogeneous cost parameters. In the corresponding numerical simulations, the cost parameters for the three firms are set to 0.2, 1.0, and 1.0, respectively. The results suggest that when the first firm—employing the gradient adjustment mechanism—has a relatively low cost parameter, complex dynamics such as chaos may arise during the process of equilibrium destabilization. This observation aligns with the findings from the previous section regarding the impact of cost asymmetry among firms.

Regarding the size of stability region of models $T_{GNA}$ and $T_{GN}$, we have the following proposition.

**Proposition 2.** *The stability region of model $T_{GNA}$ is strictly larger than that of model $T_{GN}$.*

*Proof*: It suffices to prove that

$$\frac{9(101\,l - 200)}{2(252\,l - 505)} > \sqrt{2}, \tag{38}$$

which is equivalent to

$$9(101\,l - 200) < 2\sqrt{2}(252\,l - 505) \tag{39}$$

since $252\,l - 505 < 0$. Moreover, the above inequality can be transformed to

$$(909 - 504\sqrt{2})l < (1800 - 1010\sqrt{2}), \tag{40}$$

which is true by simply verifying at $l = 0$ and $l = 1$. The proof is completed. $\qquad\square$

## 4 Game of four firms

In this section, we introduce an additional player (the fourth firm). The fourth firm adopts the LMA mechanism (see, e.g., [12,33] for additional information), which is also a boundedly rational adjustment process. In this process, the player possesses only limited knowledge of the market demand. Specifically, the firm can observe the current market price $p(t)$ and the corresponding total supply $Q(t)$, and is able to accurately estimate the slope of the price function, $p'(Q(t))$, at the point $(Q(t), p(t))$. Using this information, the firm constructs a local approximation of the demand function and anticipates the price at period $t + 1$ to be

$$p^e(t + 1) = p(Q(t)) + p'(Q(t))(Q^e(t + 1) - Q(t)), \tag{41}$$

where $Q^e(t + 1)$ represents the expected aggregate production or total market supply at period $t + 1$. Moreover, firm 4 is also assumed to employ the naive expectations of its rivals, thus

$$Q^e(t + 1) = q_1(t) + q_2(t) + q_3(t) + q_4(t + 1). \tag{42}$$

Then we have that

$$p^e(t + 1) = \frac{1}{Q(t)} - \frac{1}{Q^2(t)}(q_4(t + 1) - q_4(t)). \tag{43}$$

Accordingly, the expected profit of firm 4 becomes

$$\Pi_4^e(t+1) = p^e(t+1)q_4(t+1) - cq_4^2(t+1).\tag{44}$$

To maximize the expected profit, firm 4 chooses its output at period $t+1$ according to the first order condition

$$\frac{\partial \Pi_4^e(t+1)}{\partial q_4(t+1)} = 0,\tag{45}$$

which can be solved by

$$q_4(t+1) = \frac{2\,q_4(t) + q_1(t) + q_2(t) + q_3(t)}{2(1 + c(q_1(t) + q_2(t) + q_3(t) + q_4(t))^2)}.\tag{46}$$

Therefore, the new model can be described by the following 4-dimensional discrete dynamic system.

$$T_{GNAL}(q_1, q_2, q_3, q_4):$$
$$\begin{cases}
q_1(t+1) = q_1(t) + kq_1(t)\left[\dfrac{q_2(t) + q_3(t) + q_4(t)}{(q_1(t) + q_2(t) + q_3(t) + q_4(t))^2} - 2\,cq_1(t)\right], \\
q_2(t+1) = R_2(q_1(t), q_3(t), q_4(t)), \\
q_3(t+1) = (1-l)q_3 + lR_3(q_1(t), q_2(t), q_4(t)), \\
q_4(t+1) = \dfrac{2\,q_4(t) + q_1(t) + q_2(t) + q_3(t)}{2(1 + c(q_1(t) + q_2(t) + q_3(t) + q_4(t))^2)},
\end{cases}\tag{47}$$

where $R_2(q_1(t), q_3(t), q_4(t)) = \Phi(q_1(t) + q_3(t) + q_4(t))$ and $R_3(q_1(t), q_2(t), q_4(t)) = \Phi(q_1(t) + q_2(t) + q_4(t))$ (see Eq (7) for the expression of $\Phi$). Then, the equilibrium $(q_1^*, \dots, q_4^*)$ satisfies that

$$\begin{cases}
kq_1^*\left(\dfrac{q_2^* + q_3^* + q_4^*}{(q_1^* + q_2^* + q_3^* + q_4^*)^2} - 2\,cq_1^*\right) = 0, \\
q_1^* + q_3^* + q_4^* - 2\,cq_2^*(q_1^* + q_2^* + q_3^* + q_4^*)^2 = 0, \\
q_1^* + q_2^* + q_4^* - 2\,cq_3^*(q_1^* + q_2^* + q_3^* + q_4^*)^2 = 0, \\
q_4^* - \dfrac{2\,q_4^* + q_1^* + q_2^* + q_3^*}{2(1 + c(q_1^* + q_2^* + q_3^* + q_4^*)^2)} = 0,
\end{cases}\tag{48}$$

which is solved by two solutions

$$E_{GNAL}^1 = \left(0, \frac{1}{\sqrt{9c}}, \frac{1}{\sqrt{9c}}, \frac{1}{\sqrt{9c}}\right),$$
$$E_{GNAL}^2 = \left(\sqrt{\frac{3}{32c}}, \sqrt{\frac{3}{32c}}, \sqrt{\frac{3}{32c}}, \sqrt{\frac{3}{32c}}\right).\tag{49}$$

Hence, there exists one unique positive (Cournot–Nash) equilibrium $E_{GNAL}^2$, where the Jacobian matrix of $T_{GNAL}$ is

$$J_{GNAL}(E^2_{GNAL}) = \begin{bmatrix} 1 - 3k\sqrt{6c}/8 & -k\sqrt{6c}/24 & -k\sqrt{6c}/24 & -k\sqrt{6c}/24 \\ -1/9 & 0 & -1/9 & -1/9 \\ -l/9 & -l/9 & 1-l & -l/9 \\ -1/10 & -1/10 & -1/10 & 1/10 \end{bmatrix}. \tag{50}$$

According to Proposition 1, we have the following corollary.

**Corollary 2.** *Consider a 4-dimensional discrete dynamic system with the characteristic polynomial of its Jacobian matrix of the form*

$$A = \lambda^4 + a_3\lambda^3 + a_2\lambda^2 + a_1\lambda + a_0.$$

*An equilibrium E is locally stable if the following inequalities are satisfied at E.*

$$\begin{cases} 1 + a_3 + a_2 + a_1 + a_0 > 0, \\ 1 - a_3 + a_2 - a_1 + a_0 > 0, \\ -a_0^3 - a_0^2 a_2 + a_0 a_1 a_3 + a_0 a_3^2 - a_0^2 - a_1^2 - a_1 a_3 + a_0 + a_2 + 1 > 0, \\ a_0^3 - a_0^2 a_2 + a_0 a_1 a_3 - a_0 a_3^2 - a_0^2 + 2\,a_0 a_2 - a_1^2 + a_1 a_3 - a_0 - a_2 + 1 > 0, \\ 1 + a_0 > 0, \\ 1 - a_0 > 0. \end{cases} \tag{51}$$

For the 4-dimensional discrete dynamic system (47), one can verify that at the equilibrium $E^2_{GNAL}$, the characteristic polynomial is

$$A = \lambda^4 + \frac{\left(15k\sqrt{6} + 40l\sqrt{\frac{1}{c}} - 84\sqrt{\frac{1}{c}}\right)\lambda^3}{40\sqrt{\frac{1}{c}}} + \frac{\left(1200lk\sqrt{6} - 1365k\sqrt{6} - 3640l\sqrt{\frac{1}{c}} + 3852\sqrt{\frac{1}{c}}\right)\lambda^2}{3240\sqrt{\frac{1}{c}}}$$
$$- \frac{\left(512lk\sqrt{6} - 423k\sqrt{6} - 1128l\sqrt{\frac{1}{c}} + 756\sqrt{\frac{1}{c}}\right)\lambda}{9720\sqrt{\frac{1}{c}}}$$
$$- \frac{9\left(\frac{1}{c}\right)^{\frac{3}{2}}c^2k^2l + 9\left(\frac{1}{c}\right)^{\frac{3}{2}}c^2k^2 - 9\sqrt{\frac{1}{c}}ck^2l - 9ck^2\sqrt{\frac{1}{c}} + 128lk\sqrt{6} - 216k\sqrt{6} - 576l\sqrt{\frac{1}{c}} + 864\sqrt{\frac{1}{c}}}{77760\sqrt{\frac{1}{c}}}. \tag{52}$$

Accordingly, the above condition (51) can be reformulated into

$$\begin{aligned} CD^1_{GNAL} > 0,\ CD^2_{GNAL} > 0,\ CD^3_{GNAL} > 0, \\ CD^4_{GNAL} < 0,\ CD^5_{GNAL} < 0,\ CD^6_{GNAL} > 0, \end{aligned} \tag{53}$$

where

$$CD^1_{GNAL} = kl\sqrt{32c/3},$$

$$CD^2_{GNAL} = (512\,kl - 1017\,k)\sqrt{32c/3} - 3616\,l + 7056,$$

$$CD^3_{GNAL} = (28672\,k^3 l^3 - 1062432\,k^3 l^2 + 9180054\,k^3 l - 12603681\,k^3)(\sqrt{32c/3})^3$$
$$+ (-3777536\,k^2 l^3 + 179157888\,k^2 l^2 - 1194862752\,k^2 l + 945483840\,k^2)(\sqrt{32c/3})^2$$
$$+ (116054016\,kl^3 - 4248400896\,kl^2 - 5573546496\,kl + 13237426944\,k)\sqrt{32c/3}$$
$$- 566525952\,l^3 + 11952783360\,l^2 + 47066406912\,l - 133145026560,$$

$$CD^4_{GNAL} = (3616\,k^3 l^3 - 132966\,k^3 l^2 - 512973\,k^3 l + 1226907\,k^3)(\sqrt{32c/3})^3$$
$$+ (-472768\,k^2 l^3 + 16419744\,k^2 l^2 + 77813136\,k^2 l - 83525904\,k^2)(\sqrt{32c/3})^2$$
$$+ (-6484992\,kl^3 + 276668928\,kl^2 + 1145829888\,kl - 1868106240\,k)\sqrt{32c/3}$$
$$+ 55148544\,l^3 - 1055932416\,l^2 - 6642155520\,l,$$

$$CD^5_{GNAL} = (16\,kl - 27\,k)\sqrt{32c/3} - 96\,l - 12816,$$

$$CD^6_{GNAL} = (16\,kl - 27\,k)\sqrt{32c/3} - 96\,l + 13104,$$

(54)

To simplify the stability condition (53), it is also helpful to explore the inclusion relations of these inequalities. Bear in mind that the surfaces $CD^i_{GNAL} = 0 (i = 1, \dots, 6)$ divide the parameter set $\{(k, l, c) \mid k > 0, 1 \geq l > 0, c > 0\}$ into regions, and in each of them the signs of $CD^i_{GNAL} (i = 1, \dots, 6)$ are invariant. Similarly, we use the CAD method to select at least one sample point from each of these regions.

Table 2 lists the selected sample points and shows the verification results of the six inequalities in (53) at these sample points. It is observed that at all the sample points where $CD^2_{GNAL} > 0$ is true, the other five inequalities are also true, which implies that the stability condition (53) will be satisfied if we only have $CD^2_{GNAL} > 0$. Furthermore, it is easy to see

**Table 2. Stability conditions of $T_{GNAL}$ at selected sample points.**

| sample point of $(k, l, c)$ | $CD^1_{GNAL} > 0$ | $CD^2_{GNAL} > 0$ | $CD^3_{GNAL} > 0$ |
|---|---|---|---|
| (55/64, 109/256, 3/2) | true | true | true |
| (243/128, 109/256, 3/2) | true | false | true |
| (301/32, 109/256, 3/2) | true | false | false |
| (271/16, 109/256, 3/2) | true | false | true |
| (5725/64, 109/256, 3/2) | true | false | true |
| (20771/128, 109/256, 3/2) | true | false | true |
| (109/128, 119/128, 3/2) | true | true | true |
| (1275/256, 119/128, 3/2) | true | false | true |
| (35405/256, 119/128, 3/2) | true | false | true |
| (34413/128, 119/128, 3/2) | true | false | true |
| sample point of $(k, l, c)$ | $CD^4_{GNAL} < 0$ | $CD^5_{GNAL} < 0$ | $CD^6_{GNAL} > 0$ |
| (55/64, 109/256, 3/2) | true | true | true |
| (243/128, 109/256, 3/2) | true | true | true |
| (301/32, 109/256, 3/2) | true | true | true |
| (271/16, 109/256, 3/2) | true | true | true |
| (5725/64, 109/256, 3/2) | false | true | true |
| (20771/128, 109/256, 3/2) | false | true | false |
| (109/128, 119/128, 3/2) | true | true | true |
| (1275/256, 119/128, 3/2) | true | true | true |
| (35405/256, 119/128, 3/2) | false | true | true |
| (34413/128, 119/128, 3/2) | false | true | false |

that $CD^2_{GNAL} > 0$ is equivalent to

$$k\sqrt{c} < \frac{2\sqrt{6}(226\,l - 441)}{512\,l - 1017}.$$ (55)

Therefore, we summarize the results in the following theorem.

**Theorem 3.** *Model $T_{GNAL}$ described by* (47) *has a unique positive equilibrium*

$$\left(\sqrt{\frac{3}{32c}}, \sqrt{\frac{3}{32c}}, \sqrt{\frac{3}{32c}}, \sqrt{\frac{3}{32c}}\right),$$ (56)

*which is locally stable if*

$$k\sqrt{c} < \frac{2\sqrt{6}(226\,l - 441)}{512\,l - 1017}.$$ (57)

Fig 3 reports bifurcation diagrams of model $T_{GNAL}$, where numerical simulations are conducted by choosing the initial state to be $(q_1(0), \ldots, q_4(0)) = (0.1, \ldots, 0.1)$. In Fig 3(a), the one-dimensional bifurcation diagram with respect to $k$ is depicted by fixing $c = 1.0$ and $l = 0.5$. We mark the diagram against $q_1, \ldots, q_4$ in red, blue, green, and black, respectively. It can be observed that the equilibrium transitions to 2-cycle orbits when $k = 2.110552764$ and the trajectories diverge to $\infty$ when $k = 2.226130653$. It is also discovered that the amplitude of the 2-cycle orbits of firm 1 is much larger than that of its rivals. Fig 3(b) depicts the one-dimensional bifurcation diagram with respect to $c$ by fixing $k = 1.0$ and $l = 0.5$. One can see that the dynamics of Fig 3(b) are much more complex than those of Fig 3(a). In Fig 3(b), chaos takes place through a series of period-doubling bifurcations, whereas in Fig 3(a), there exist only equilibria and 2-cycle orbits. Furthermore, we discover that the amplitude of firm 1's output is much larger than its competitors. Similarly, it is found that an increase in *korc* has a destabilizing effect.

If we vary the value of $l$, the same phenomena of the non-existence of chaotic dynamics can be found (see Fig 3(c) for details). Furthermore, we provide Fig 3(d) to explore the situation of the firms possessing different cost parameters. Specifically, in Fig 3(d), we plot the two-dimensional bifurcation diagram with respect to $l$ and $k$ by setting the cost parameters of the four firms to be 0.5, 1.4, 2.0 and 0.5, respectively. Readers can refer to [26] for more information regarding two-dimensional bifurcation diagrams. It can be concluded that complex dynamics such as chaos may appear if the cost parameter of the first firm adopting the gradient adjustment mechanism is relatively small enough.

Then, regarding the size of the stability region of model $T_{GNAL}$, we have the following result.

**Proposition 3.** *The stability region of model $T_{GNAL}$ is strictly larger than that of model $T_{GNA}$.*

*Proof*: It suffices to prove that

$$\frac{9(101\,l - 200)}{2(252\,l - 505)} < \frac{2\sqrt{6}(226\,l - 441)}{512\,l - 1017},$$ (58)

which is equivalent to

$$9(101\,l - 200)(512\,l - 1017) < 4\sqrt{6}(252\,l - 505)(226\,l - 441),$$ (59)

and further to

$$(-227808\sqrt{6} + 465408)l^2 + (901048\sqrt{6} - 1846053)l - 890820\sqrt{6} + 1830600 < 0.$$ (60)

The above inequality is satisfied for $0 < l \leq 1$ since the left part has a negative leading coefficient and both of it roots are greater than 1, which completes the proof. □

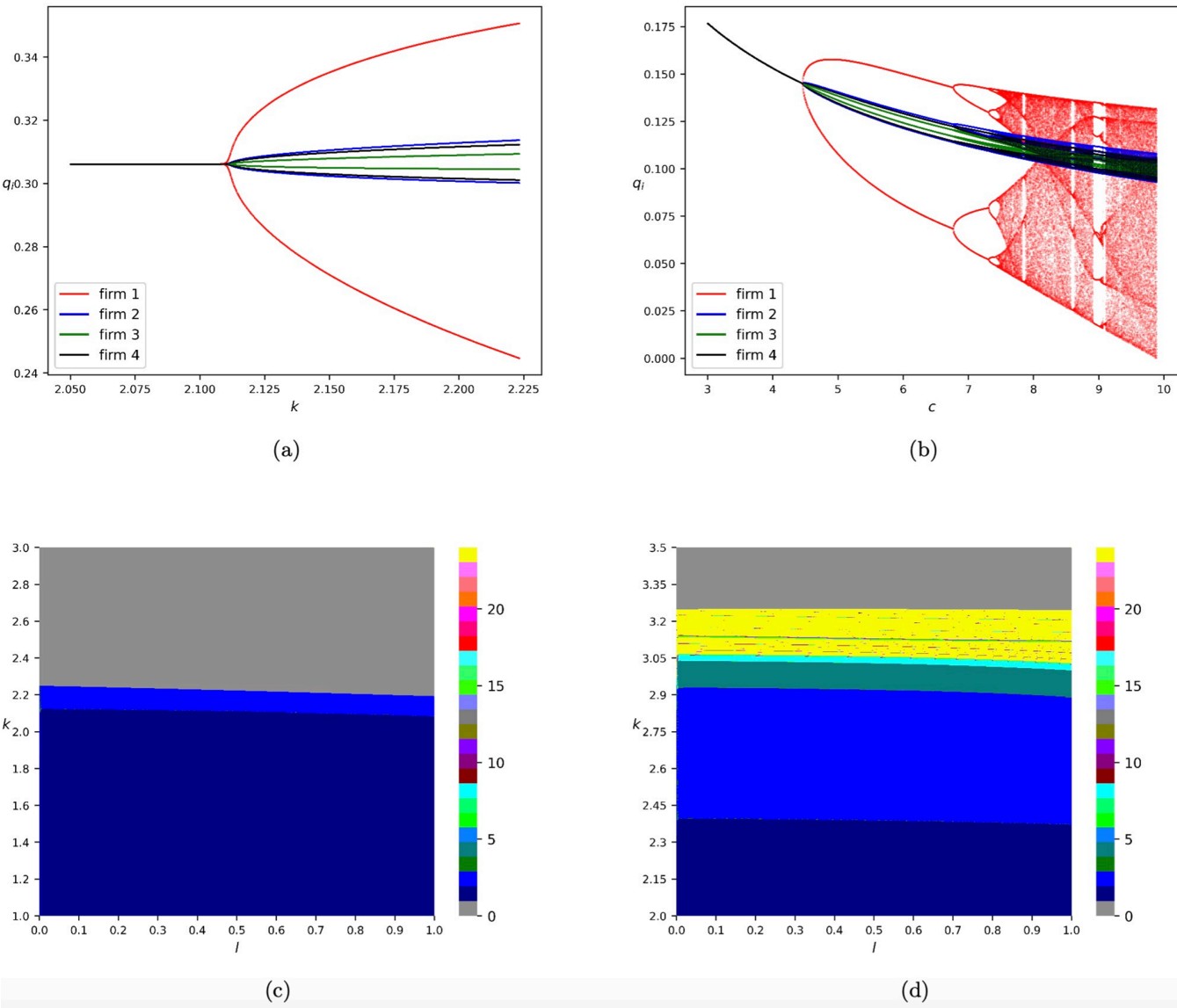

**Fig 3. Bifurcation diagrams of model $T_{GNAL}$.** (a) One-dimensional bifurcation diagram with respect to $k$ by fixing $c = 1.0$ and $l = 0.5$. (b) One-dimensional bifurcation diagram with respect to $c$ by fixing $k = 1.0$ and $l = 0.5$. The diagram against $q_1, \dots, q_4$ is marked in red, blue, green, and black, respectively. The diagram against $q_1, \dots, q_4$ is marked in red, blue, green, and black, respectively. (c) Two-dimensional bifurcation diagram with respect to $l$ and $k$ by fixing $c = 1.0$. (d) Two-dimensional bifurcation diagram with respect to $l$ and $k$ if we set the cost parameters of the four firms to be 0.5, 1.4, 2.0, and 0.5, respectively. All the numerical simulations are conducted by choosing the initial iteration state to be $(q_1(0), \dots, q_4(0)) = (0.1, \dots, 0.1)$.

## 5 Game of five firms

Finally, we incorporate a special firm into the model—a *rational player*. Unlike firm 2, this rational player has full knowledge of the price function's form and complete information about its competitors' decisions. In contrast, firm 2 lacks information about its rivals' current production plans and therefore naively assumes that they will produce the same quantities as in the previous period. As a result, firm 2's expected profit at period $t + 1$ is given by

$$\Pi_2^e(t+1) = \frac{q_2(t+1)}{q_1(t) + q_2(t+1) + q_3(t) + q_4(t) + q_5(t)} - cq_2^2(t+1). \tag{61}$$

In comparison, firm 5 has complete information and knows exactly all its rivals' production plans. Hence, the expected profit of firm 5 should be the real profit, i.e.,

$$\Pi_5^e(t+1) = \Pi_5(t+1) =$$
$$\frac{q_5(t+1)}{q_1(t+1) + q_2(t+1) + q_3(t+1) + q_4(t+1) + q_5(t+1)} - cq_5^2(t+1). \tag{62}$$

To maximize its profit, firm 5 decides its output at period $t+1$ by solving the first condition $\partial \Pi_5(t+1)/\partial q_5(t+1) = 0 \, for \, q_5(t+1)$. Denote the solution as

$$q_5(t+1) = R_5(q_1(t+1), q_2(t+1), q_3(t+1), q_4(t+1)). \tag{63}$$

Actually, we have

$$R_5(q_1(t+1), q_2(t+1), q_3(t+1), q_4(t+1)) = \Phi(q_1(t+1) + q_2(t+1) + q_3(t+1) + q_4(t+1)). \tag{64}$$

The expression of $\Phi$ can be found in Eq (7). Therefore, the model can be formulated as the 5-dimensional iteration map

$$T_{GNALR}(q_1, q_2, q_3, q_4, q_5):$$
$$\begin{cases} q_1(t+1) = q_1(t) + kq_1(t)\left[\dfrac{q_2(t) + q_3(t) + q_4(t) + q_5(t)}{(q_1(t) + q_2(t) + q_3(t) + q_4(t) + q_5(t))^2} - 2cq_1(t)\right], \\ q_2(t+1) = R_2(q_1(t), q_3(t), q_4(t), q_5(t)), \\ q_3(t+1) = (1-l)q_3(t) + lR_3(q_1(t), q_2(t), q_4(t), q_5(t)), \\ q_4(t+1) = \dfrac{2q_4(t) + q_1(t) + q_2(t) + q_3(t) + q_5(t)}{2(1 + c(q_1(t) + q_2(t) + q_3(t) + q_4(t) + q_5(t))^2)}, \\ q_5(t+1) = R_5(q_1(t+1), q_2(t+1), q_3(t+1), q_4(t+1)), \end{cases} \tag{65}$$

where

$$\begin{aligned} R_2(q_1(t), q_3(t), q_4(t), q_5(t)) &= \Phi(q_1(t) + q_3(t) + q_4(t) + q_5(t)), \\ R_3(q_1(t), q_2(t), q_4(t), q_5(t)) &= \Phi(q_1(t) + q_2(t) + q_4(t) + q_5(t)), \\ R_5(q_1(t+1), q_2(t+1), q_3(t+1), q_4(t+1)) &= \\ \Phi(q_1(t+1) + q_2(t+1) + q_3(t+1) + q_4(t+1)). \end{aligned} \tag{66}$$

Therefore, the equilibrium $(q_1^*, \ldots, q_5^*)$ are described by

$$\begin{cases} kq_1^*\left(\dfrac{q_2^* + q_3^* + q_4^* + q_5^*}{(q_1^* + q_2^* + q_3^* + q_4^* + q_5^*)^2} - 2cq_1\right) = 0, \\ q_1^* + q_3^* + q_4^* + q_5^* - 2cq_2^*(q_1^* + q_2^* + q_3^* + q_4^* + q_5^*)^2 = 0, \\ q_1^* + q_2^* + q_4^* + q_5^* - 2cq_3^*(q_1^* + q_2^* + q_3^* + q_4^* + q_5^*)^2 = 0, \\ q_4^* - \dfrac{2q_4^* + q_1^* + q_2^* + q_3^* + q_5^*}{2(1 + c(q_1^* + q_2^* + q_3^* + q_4^* + q_5^*)^2)} = 0, \\ q_1^* + q_2^* + q_3^* + q_4^* - 2cq_5^*(q_1^* + q_2^* + q_3^* + q_4^* + q_5^*)^2 = 0, \end{cases} \tag{67}$$

of which we can compute two solutions

$$E^1_{GNALR} = \left(0, \sqrt{\frac{3}{32c}}, \sqrt{\frac{3}{32c}}, \sqrt{\frac{3}{32c}}, \sqrt{\frac{3}{32c}}\right),$$

$$E^2_{GNALR} = \left(\sqrt{\frac{2}{25c}}, \sqrt{\frac{2}{25c}}, \sqrt{\frac{2}{25c}}, \sqrt{\frac{2}{25c}}, \sqrt{\frac{2}{25c}}\right). \tag{68}$$

For the sake of simplicity, we denote the first and fourth equations in (65) to be

$$q_1(t+1) = G_1(q_1(t), q_2(t), q_3(t), q_4(t), q_5(t)), \tag{69}$$

and

$$q_4(t+1) = L_4(q_1(t), q_2(t), q_3(t), q_4(t), q_5(t)), \tag{70}$$

respectively. From the last equation in map (65), we have

$$q_5(t) = R_5(q_1(t), q_2(t), q_3(t), q_4(t)),$$

which can be used to replace $q_5(t)$ in the first 4 equations in map (65). Accordingly, map (65) can be reformulated into the following equivalent 4-dimensional map.

$$T_{GNALR}(q_1, q_2, q_3, q_4):$$
$$\begin{cases} q_1(t+1) = G_1(q_1(t), q_2(t), q_3(t), q_4(t), R_5(q_1(t), q_2(t), q_3(t), q_4(t))), \\ q_2(t+1) = R_2(q_1(t), q_3(t), q_4(t), R_5(q_1(t), q_2(t), q_3(t), q_4(t))), \\ q_3(t+1) = (1-l)q_3(t) + lR_3(q_1(t), q_2(t), q_4(t), R_5(q_1(t), q_2(t), q_3(t), q_4(t))), \\ q_4(t+1) = L_4(q_1(t), q_2(t), q_3(t), q_4(t), R_5(q_1(t), q_2(t), q_3(t), q_4(t))). \end{cases} \tag{71}$$

Hence, the analysis of the local stability is transformed into the investigation of the Jacobian matrix

$$J_{GNALR} = \begin{bmatrix} \frac{\partial q_1(t+1)}{\partial q_1(t)} & \frac{\partial q_1(t+1)}{\partial q_2(t)} & \frac{\partial q_1(t+1)}{\partial q_3(t)} & \frac{\partial q_1(t+1)}{\partial q_4(t)} \\ \frac{\partial q_2(t+1)}{\partial q_1(t)} & \frac{\partial q_2(t+1)}{\partial q_2(t)} & \frac{\partial q_2(t+1)}{\partial q_3(t)} & \frac{\partial q_2(t+1)}{\partial q_4(t)} \\ \frac{\partial q_3(t+1)}{\partial q_1(t)} & \frac{\partial q_3(t+1)}{\partial q_2(t)} & \frac{\partial q_3(t+1)}{\partial q_3(t)} & \frac{\partial q_3(t+1)}{\partial q_4(t)} \\ \frac{\partial q_4(t+1)}{\partial q_1(t)} & \frac{\partial q_4(t+1)}{\partial q_2(t)} & \frac{\partial q_4(t+1)}{\partial q_3(t)} & \frac{\partial q_4(t+1)}{\partial q_4(t)} \end{bmatrix}, \tag{72}$$

where

$$\frac{\partial q_1(t+1)}{\partial q_i(t)} = \frac{\partial G_1}{\partial q_i} + \frac{\partial G_1}{\partial R_5}\frac{\partial R_5}{\partial q_i}, \quad i = 1, 2, 3, 4,$$

$$\frac{\partial q_2(t+1)}{\partial q_i(t)} = \frac{\partial R_2}{\partial q_i} + l\frac{\partial R_2}{\partial R_5}\frac{\partial R_5}{\partial q_i}, \quad i = 1, 3, 4,$$

$$\frac{\partial q_2(t+1)}{\partial q_2(t)} = \frac{\partial R_2}{\partial R_5}\frac{\partial R_5}{\partial q_2},$$

$$\frac{\partial q_3(t+1)}{\partial q_i(t)} = l\frac{\partial R_3}{\partial q_i} + l\frac{\partial R_3}{\partial R_5}\frac{\partial R_5}{\partial q_i}, \quad i = 1, 2, 4, \tag{73}$$

$$\frac{\partial q_3(t+1)}{\partial q_3(t)} = (1-l) + l\frac{\partial R_3}{\partial R_5}\frac{\partial R_5}{\partial q_3},$$

$$\frac{\partial q_4(t+1)}{\partial q_i(t)} = \frac{\partial L_4}{\partial q_i} + \frac{\partial L_4}{\partial R_5}\frac{\partial R_5}{\partial q_i}, \quad i = 1, 2, 3, 4.$$

Likewise, we focus on the positive (Cournot–Nash) equilibrium $E^2_{GNALR}$, where the Jacobian matrix $\boldsymbol{J}_{GNALR}$ becomes

$$\boldsymbol{J}_{GNALR}(E^2_{GNALR}) =$$
$$\begin{bmatrix} 1 - 31\,k\sqrt{2c}/56 & -3\,k\sqrt{2c}/56 & -3\,k\sqrt{2c}/56 & -3\,k\sqrt{2c}/56 \\ -75/784 & 9/784 & -75/784 & -75/784 \\ 0 & 0 & 1 - 25\,l/28 & 0 \\ -5/56 & -5/56 & -5/56 & 13/168 \end{bmatrix}. \tag{74}$$

The characteristic polynomial is quite complicated, which is not reported here due to space limitations. But, according to Corollary 2, we can conclude that the equilibrium $E^2_{GNALR}$ is locally stable if

$$CD^1_{GNALR} > 0, \ CD^2_{GNALR} > 0, \ CD^3_{GNALR} < 0,$$
$$CD^4_{GNALR} < 0, \ CD^5_{GNALR} < 0, \ CD^6_{GNALR} > 0, \tag{75}$$

where

$$CD^1_{GNALR} = kl\sqrt{25c/2},$$
$$CD^2_{GNALR} = (25\,l - 56)(5737\,k\sqrt{25c/2} - 50860),$$
$$\begin{aligned} CD^3_{GNALR} = & (3934321875\,k^3l^3 - 104905111500\,k^3l^2 + 1172129631120\,k^3l \\ & - 1186719653952\,k^3)(\sqrt{25c/2})^3 + (-439562531250\,k^2l^3 + 19054516460000\,k^2l^2 \\ & - 144796527937600\,k^2l + 134072666053760\,k^2)(\sqrt{25c/2})^2 + (19706242500000\,kl^3 \\ & - 579386747450000\,kl^2 - 1721529608680000\,kl + 3133067852544000\,k)\sqrt{25c/2} \\ & - 113004562500000\,l^3 + 1975821995000000\,l^2 + 12875890524000000\,l \\ & - 37485773024000000, \end{aligned} \tag{76}$$
$$\begin{aligned} CD^4_{GNALR} = & (9423\,k^2(\sqrt{25c/2})^2 - 981050\,k\sqrt{25c/2} - 33575000)((3375\,kl^3 - 89180\,kl^2 \\ & - 629552\,kl + 812224\,k)\sqrt{25c/2} \\ & - 22500\,l^3 + 343000\,l^2 + 3332000\,l) \end{aligned}$$
$$CD^5_{GNALR} = (225\,kl - 252\,k)\sqrt{25c/2} - 1500\,l - 217840,$$
$$CD^6_{GNALR} = (225\,kl - 252\,k)\sqrt{25c/2} - 1500\,l + 221200.$$

In Table 3, we list all the selected sample points generated by the CAD method such that there exists at least one point in each region divided by the surfaces $CD^i_{GNALR} = 0$, $i = 1, \dots, 6$. The six inequalities in (75) are verified at these sample points one by one, and the results are also reported in Table 3. From Table 3, one can see that the equilibrium $E^2_{GNALR}$ is locally stable if $CD^2_{GNALR} > 0$. Therefore, we have the following theorem.

**Theorem 4.** *Model* $T_{GNALR}$ *described by* (65) *has a unique positive equilibrium*

$$\left( \sqrt{\frac{2}{25c}}, \sqrt{\frac{2}{25c}}, \sqrt{\frac{2}{25c}}, \sqrt{\frac{2}{25c}}, \sqrt{\frac{2}{25c}} \right), \tag{77}$$

*which is locally stable if*

$$k\sqrt{c} < \frac{10172\sqrt{2}}{5737}. \tag{78}$$

**Table 3. Stability conditions of $T_{GNALR}$ at selected sample points.**

| sample point of $(k, l, c)$ | $CD^1_{GNALR} > 0$ | $CD^2_{GNALR} > 0$ | $CD^3_{GNALR} < 0$ |
|---|---|---|---|
| (453/256, 61/128, 1/2) | true | true | true |
| (1007/256, 61/128, 1/2) | true | false | true |
| (7183/256, 61/128, 1/2) | true | false | false |
| (6675/128, 61/128, 1/2) | true | false | true |
| (10587/32, 61/128, 1/2) | true | false | true |
| (9755/16, 61/128, 1/2) | true | false | true |
| (453/256, 251/256, 1/2) | true | true | true |
| (1567/256, 251/256, 1/2) | true | false | true |
| (225/8, 251/256, 1/2) | true | false | false |
| (12807/256, 251/256, 1/2) | true | false | true |
| (91267/64, 251/256, 1/2) | true | false | true |
| (89603/32, 251/256, 1/2) | true | false | true |
| sample point of $(k, l, c)$ | $CD^4_{GNALR} < 0$ | $CD^5_{GNALR} < 0$ | $CD^6_{GNALR} > 0$ |
| (453/256, 61/128, 1/2) | true | true | true |
| (1007/256, 61/128, 1/2) | true | true | true |
| (7183/256, 61/128, 1/2) | true | true | true |
| (6675/128, 61/128, 1/2) | true | true | true |
| (10587/32, 61/128, 1/2) | false | true | true |
| (9755/16, 61/128, 1/2) | false | true | false |
| (453/256, 251/256, 1/2) | true | true | true |
| (1567/256, 251/256, 1/2) | true | true | true |
| (225/8, 251/256, 1/2) | true | true | true |
| (12807/256, 251/256, 1/2) | true | false | true |
| (91267/64, 251/256, 1/2) | false | true | true |
| (89603/32, 251/256, 1/2) | false | true | false |

Fig 4 reports the bifurcation diagrams of the equivalent 4-dimensional map (71) of model $T_{GNALR}$, where we choose the initial iteration state to be $(q_1(0), \dots, q_4(0)) = (0.1, \dots, 0.1)$. Specifically, Fig 4(a) depicts the one-dimensional bifurcation diagram of map (71) with respect to $k$ by fixing $c = 1.0$ and $l = 0.5$. One can see that the dynamics of model $T_{GNALR}$ are more complex than those of models $T_{GNA}$ and $T_{GNAL}$ when the cost parameters of the involved firms are identical in the sense that 4-cycle orbits appear in Fig 4(a). It is also observed that the amplitude of the periodic orbits of the first player is much larger than its competitors. In Fig 4(b), we report the one-dimensional bifurcation diagram of map (71) with respect to $c$ by fixing $k = 1.0$ and $l = 0.5$. Compared to the diagram with respect to $k$ (Fig 4(a)), the dynamics shown by the diagram with respect to $c$ (Fig 4(b)) are much more complex in the sense that chaos may appear if the value of $c$ is large enough. One can also find that a decrease in $k$ or $c$ leads to the stabilization of the model.

One can refer to Fig 4(c) to explore the situation if we vary the adaptive proportion parameter $l$. We find that if the value of $l$ is sufficiently small, then 8-cycle orbits may appear. However, chaotic dynamics can not occur in the case of identical cost parameters. As an example of distinct cost parameters, Fig 4(d) reports the two-dimensional bifurcation diagram with respect to $l$ and $k$ by setting the cost parameters of the five firms to be 1.0, 2.0, 1.4, 1.6, and 1.8, respectively. We conclude that complex dynamics, such as chaos and periodic orbits with high orders, of the game may occur if the costs of the involved companies are sufficiently different.

Furthermore, the following result is acquired.

**Proposition 4.** *The stability region of model $T_{GNALR}$ is strictly larger than that of model $T_{GNAL}$.*

*Proof*: It suffices to prove that

$$\frac{2\sqrt{6}(226 l - 441)}{512 l - 1017} < \frac{10172\sqrt{2}}{5737},$$ 
(79)

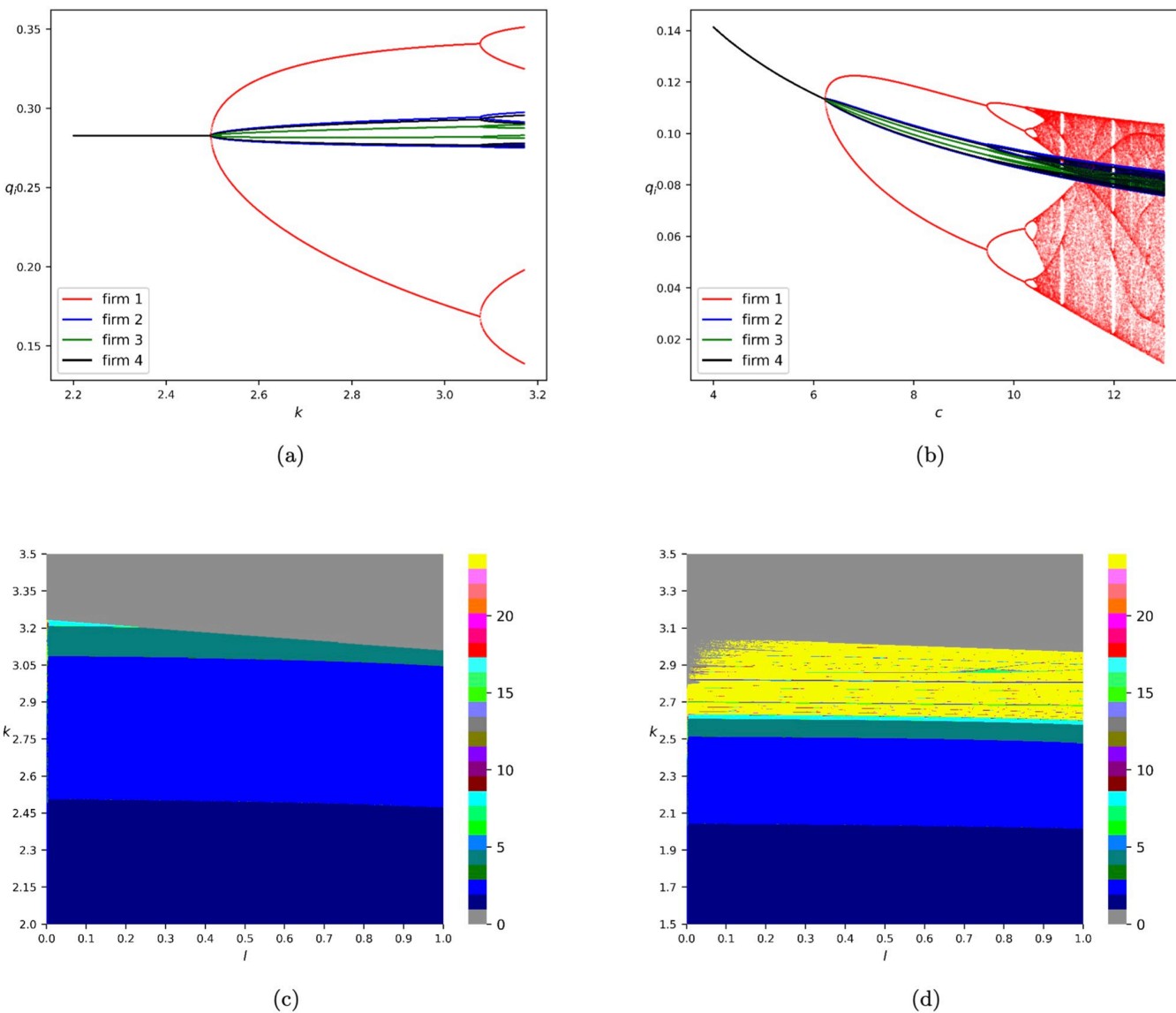

**Fig 4**. **Bifurcation diagrams of the equivalent 4-dimensional map** (71) **of model** $T_{GNALR}$. (a) One-dimensional bifurcation diagram with respect to $k$ by fixing $c = 1.0$ and $l = 0.5$. The diagram against $q_1, \dots, q_4$ is marked in red, blue, green, and black, respectively. (b) One-dimensional bifurcation diagram with respect to $c$ by fixing $k = 1.0$ and $l = 0.5$. The diagram against $q_1, \dots, q_4$ is marked in red, blue, green, and black, respectively. (c) Two-dimensional bifurcation diagram with respect to $l$ and $k$ by fixing $c = 1.0$. (d) Two-dimensional bifurcation diagram with respect to $l$ and $k$ if we set the cost parameters of the five firms to be 1.0, 2.0, 1.4, 1.6, and 1.8, respectively. All the numerical simulations are conducted by choosing the initial iteration state of map (71) to be $(q_1(0), \dots, q_4(0)) = (0.1, \dots, 0.1)$.

i.e.,

$$10172\sqrt{2}(1017 - 512\,l) - 5737 \times 2\sqrt{6}(441 - 226\,l) > 0, \tag{80}$$

which is true by checking at $l = 0$ and $l = 1$. The proof is completed. □

## 6 Discussions

In the literature on oligopolistic games, a homogeneous oligopoly typically refers to a setting where all firms adjust their output using identical dynamic strategies. In contrast, a heterogeneous oligopoly assumes that firms follow different decision-making rules to determine their output levels. The heterogeneous model is considered more realistic, as it is uncommon for firms in real-world economies to behave identically, given the wide variety of strategic possibilities.

This notion is echoed in ecology, where the competitive exclusion principle states that species occupying identical niches cannot coexist indefinitely. The same principle applies in economics: firms with identical strategies are unlikely to survive side by side over the long term. Instead, firms with diverse business strategies—arising from differences in risk preferences, access to information, or competitive positioning—often coexist within the same industry. However, in certain sectors, such as the Internet industry, homogeneous firms may coexist only temporarily before the system reaches a stable equilibrium.

In this regard, our study focuses on the stability of several oligopoly games by combining players adopting different output adjustment mechanisms. Intuitively, we can summarize the findings of Propositions 2, 3, and 4 in Fig 5, where we depict the stability regions of all the models considered in the paper. The Cournot–Nash equilibrium of model $T_{GN}$ is locally stable if the parameters take values from the red region. The Cournot–Nash equilibrium of model $T_{GNA}$ is locally stable if the parameters take values from the cyan and red regions. The region for the stability of model $T_{GNAL}$ is the union of the stability region of $T_{GNA}$ and the yellow one. In addition, the stability region of model $T_{GNALR}$ combines the stability region of $T_{GNAL}$ and the green one. Tramontana et al. [17] discussed the same issue as our study by considering an oligopoly market with heterogeneous firms and examined three kinds of oligopoly models: duopoly, triopoly, and quadropoly. Differently, all firms in their models are assumed to possess linear cost functions. Their results indicate that an increase in the number of heterogeneous firms may enhance the stability of the model equilibrium. However, compared to the analog [17, Fig 1] in the case of linear costs, the result shown by Fig 5 in this paper seems more evident in the sense that the stability regions of games with fewer firms are strictly contained in those with more firms.

As previously noted, in dynamic Cournot games, an increase in the number of homogeneous firms tends to destabilize the system. From an economic perspective, however, this is somewhat counterintuitive, as real-world markets often support the coexistence of many competing firms. The main contribution of this study is the finding that increasing the number of heterogeneous firms can, in fact, enhance the stability of dynamic Cournot games. This result aligns with the

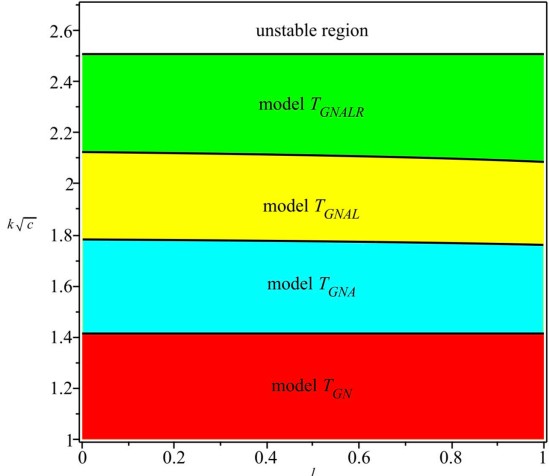

**Fig 5**. **The stability regions of all the models considered in this paper.**

expectation that firms employing diverse strategies are more likely to coexist and remain stable in competitive markets, while those with identical strategies are more vulnerable to elimination and market volatility.

It is also worth noting that the expansion of the stability region may depend on the order in which firms enter the market. Nonetheless, our findings suggest that an economy can maintain stability even as a wide variety of firms with different strategic behaviors enter the industry, supporting the intuition that heterogeneity promotes resilience in complex economic systems.

## 7 Concluding remarks

In this study, we explored oligopolistic markets with varying numbers of heterogeneous firms, focusing on models characterized by isoelastic demand and quadratic cost functions. The models encompass a series of oligopoly scenarios in which the number of participating firms increases incrementally from two to five. By progressively introducing additional heterogeneous firms, we constructed new games and analyzed how the size of the stability region evolves with firm entry. For each model, we derived analytical conditions for the local stability of the Cournot–Nash equilibrium.

A key contribution of this work is the finding that the stability regions in heterogeneous Cournot models may expand as the number of firms increases, contrary to classical results for homogeneous Cournot models, in which additional firms typically lead to instability. Fig 5 illustrates these dynamics in greater detail. Numerical simulations were also conducted to investigate the complex behaviors that emerge when equilibrium loses stability, revealing phenomena such as periodic cycles and chaotic orbits. Furthermore, simulations with distinct cost parameters revealed that heterogeneity in cost structures introduces greater complexity than the case of identical costs.

To derive the stability conditions, we employed a computational method known as CAD, which allows for the exact symbolic analysis of semi-algebraic sets. Unlike numerical approaches, the CAD method yields exact and error-free results, making it particularly suitable for proving economic theorems rigorously. We applied this method to systematically and automatically determine the local stability conditions of the models considered, offering clearer and more accessible proofs compared to the manual, pencil-and-paper approach used by Tramontana et al. [17]. Readers can see that the CAD method offers clear advantages in analytical exploration. Beyond industrial organization theory, it also holds potential applications in algorithmic game theory and theoretical macroeconomic analysis.

Our results suggest that the stability of the economic system can improve with the entry of additional heterogeneous firms, although this may depend on the order in which firms enter the market. Nevertheless, the findings support the intuitive expectation that economies can remain stable even in the presence of a large number of strategically diverse firms. Investigating the specific impact of firm entry order on system stability remains an avenue for future research.

## Acknowledgments

The authors are grateful to the anonymous referees for their helpful comments. An earlier and preliminary draft of this article can be found at: https://arxiv.org/abs/2112.13844.

## Author contributions

**Conceptualization:** Ruirui Hou.

**Formal analysis:** Xiaoliang Li.

**Investigation:** Wenshuang Wan.

**Methodology:** Xiaoliang Li.

**Validation:** Xiaoliang Li, Wenshuang Wan.

**Visualization:** Ruirui Hou, Xiaoliang Li.

**Writing – original draft:** Xiaoliang Li.

**Writing – review & editing:** Ruirui Hou, Wenshuang Wan.

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
