## [Decision Letter · Decision Letter 0]

28 Apr 2025

PONE-D-25-18321Stability analysis of heterogeneous oligopoly games of increasing players: a computational approachPLOS ONE

Dear Dr. Li,

Thank you for submitting your manuscript to PLOS ONE. After careful consideration, we feel that it has merit but does not fully meet PLOS ONE’s publication criteria as it currently stands. Therefore, we invite you to submit a revised version of the manuscript that addresses the points raised during the review process.

We look forward to receiving your revised manuscript.

Kind regards,

Pu-yan Nie

Academic Editor

PLOS ONE

Journal Requirements:

https://www.mdpi.com/2504-3110/6/8/459?

In your revision ensure you cite all your sources (including your own works), and quote or rephrase any duplicated text outside the methods section. Further consideration is dependent on these concerns being addressed.

The authors are grateful to the anonymous referees for their helpful comments. The first author is partially supported by Key Laboratory of Interdisciplinary Research of Computation and Economics (Shanghai University of Finance and Economics) and Innovation Team Project of Guangdong Colleges and Universities (Grant No. 2024WCXTD019). An earlier and preliminary draft of this article can be found at: https://arxiv.org/pdf/ 2112.13844.

The second author is partially supported by Key Laboratory of Interdisciplinary Research of Computation and Economics (Shanghai University of Finance and Economics) and Innovation Team Project of Guangdong Colleges and Universities (Grant No. 2024WCXTD019).

5. We note that your Data Availability Statement is currently as follows: All relevant data are within the manuscript and its Supporting Information files.

Reviewers' comments:

Reviewer's Responses to Questions

**Comments to the Author**

1. Is the manuscript technically sound, and do the data support the conclusions?

Reviewer #1: Yes

Reviewer #2: Yes

2. Has the statistical analysis been performed appropriately and rigorously?

Reviewer #1: N/A

Reviewer #2: N/A

3. Have the authors made all data underlying the findings in their manuscript fully available?

Reviewer #1: Yes

Reviewer #2: Yes

4. Is the manuscript presented in an intelligible fashion and written in standard English?

Reviewer #1: Yes

Reviewer #2: Yes

5. Review Comments to the Author

Reviewer #1: This study has examined oligopolistic competitive markets with varying numbers of heterogeneous firms, focusing on the isoelastic demand and quadratic cost functions. The models considered include a range of oligopoly scenarios where the number of participating firms increases from two to five. By gradually adding more heterogeneous enterprises, new games have been created. The analytical conditions for the local stability of the Cournot-Nash equilibrium of each model under consideration have been obtained. Numerical simulations have been conducted to explore the complex dynamics, such as periodic and chaotic orbits, when the game equilibrium loses its stability. Furthermore, numerical simulations have also investigated the case of distinct cost parameters. A computational tool called the CAD method has been employed to select sample points of semi-algebraic sets. The CAD method has been applied to systematically and automatically determine the local stability conditions for the equilibrium of the models considered in this manuscript, offering more comprehensible proofs than the pencil-and-paper approach. I have the following concerns.

1. The manuscript needs extensive revision for language and grammar.

2. The manuscript’s short title in the table on the first page is the same as the complete title. It should be shorter than the original title.

3. The second author has two affiliations, which should be marked with different letters (b and c).

4. All in-text mathematical expressions should be written in italics.

5. After introducing an abbreviation by its complete form in its first use, all complete forms should be replaced by the abbreviation.

6. Several references are not explained adequately in the sixth paragraph of the introduction. Each reference should be described at least in an individual sentence.

7. The equations in the introduction and their associated text should be moved to section 2.

8. All equations should be numbered throughout the manuscript.

9. What does ‘isoelastic demand function’ mean, and how can it be understood that a demand function is isoelastic or not?

10. At the bottom of page 8, in the equation before ‘which implies that’, the last term should be multiplied by ‘(q1(t)+q2(t+1))’. In the paragraph below Equation (9), there is an extra ‘(’ on the right-hand side of the expression related to ‘R3’. The ‘CD_{GNA}’ on page 20 should be corrected as ‘CD_{GNAL}’. There are two ‘(a)’ in the caption of Figure 3, and one of them should be deleted. In the partial derivatives on page 26, ‘q5’ in the denominator of fractions should be corrected as ‘R5’. In the last paragraph of page 27, the second ‘c=1’ should be corrected as ‘k=1’.

11. In all bifurcation diagrams, the x-axis should be limited to the bifurcation parameter interval, and the empty regions should be removed. Also, the bifurcation parameter should change with smaller steps to avoid discontinuities at the bifurcation points. All bifurcation parameters should be replotted.

12. All color bars should have titles, and the colors associated with chaotic and unbounded behaviors should be marked alongside different periods.

13. The ‘Acknowledgement’ and ‘Data Availability Statement’ at the end of the manuscript do not agree with those in the table at the beginning of the manuscript. They should be unified.

14. The ‘Future Works’ and ‘Authors Contributions’ sections should be added to the manuscript.

Reviewer #2: Dear Authors,

I read this manuscript and found it very useful. The authors need to consider the following comments.

1. You need to cite the number of the reference and not the year of the publication. Please the introduction part.

2. In page 8, the second equation ( q*_1-2cq*_2(q*_1+q*_2)=0 ) in the third system should be revised.

3. In Fig. 1, you considered a fixed value for k (k=1.0). What happen for other values?

4. You need to mention that (c>0) in all the equilibrium points.

5. The introduction section should be more informative. The relative papers are highly recommended to improve this section. I suggest some papers such as:

https://dx.doi.org/10.22436/jmcs.036.03.05, https://doi.org/10.1142/S0217984925501039, https://doi.org/10.28924/2291-8639-21-2023-131, https://doi.org/10.3390/fractalfract7050344

Best wishes

6. PLOS authors have the option to publish the peer review history of their article (what does this mean?). If published, this will include your full peer review and any attached files.

Reviewer #1: No

Reviewer #2: No

---

## [Author Response · Author response to Decision Letter 1]

23 May 2025

Please see the attachment for our response.

---

## [Decision Letter · Decision Letter 1]

29 Sep 2025

PONE-D-25-18321R1Stability analysis of heterogeneous oligopoly games of increasing players: A computational approachPLOS ONE

Dear Dr. Li,

Thank you for submitting your manuscript to PLOS ONE. After careful consideration, we feel that it has merit but does not fully meet PLOS ONE’s publication criteria as it currently stands. Therefore, we invite you to submit a revised version of the manuscript that addresses the points raised during the review process.

We look forward to receiving your revised manuscript.

Kind regards,

Pu-yan Nie

Academic Editor

PLOS ONE

Journal Requirements:

Reviewers' comments:

Reviewer's Responses to Questions

**Comments to the Author**

1. If the authors have adequately addressed your comments raised in a previous round of review and you feel that this manuscript is now acceptable for publication, you may indicate that here to bypass the “Comments to the Author” section, enter your conflict of interest statement in the “Confidential to Editor” section, and submit your "Accept" recommendation.

Reviewer #1: All comments have been addressed

Reviewer #2: All comments have been addressed

Reviewer #3: All comments have been addressed

2. Is the manuscript technically sound, and do the data support the conclusions?

Reviewer #1: Yes

Reviewer #2: Yes

Reviewer #3: Yes

3. Has the statistical analysis been performed appropriately and rigorously?

Reviewer #1: N/A

Reviewer #2: N/A

Reviewer #3: Yes

4. Have the authors made all data underlying the findings in their manuscript fully available?

Reviewer #1: Yes

Reviewer #2: Yes

Reviewer #3: Yes

5. Is the manuscript presented in an intelligible fashion and written in standard English?

Reviewer #1: Yes

Reviewer #2: Yes

Reviewer #3: Yes

6. Review Comments to the Author

Reviewer #1: (No Response)

Reviewer #2: Dear Authors,

The provided comments have been taken into consideration. The paper is now fine.

Best wishes

Reviewer

Reviewer #3: This paper investigates Cournot competition with heterogeneous firms under isoelastic demand and quadratic cost functions. By sequentially introducing additional heterogeneous firms, the authors analyze how the size of the stability region evolves as the number of firms increases. The paper makes use of the cylindrical algebraic decomposition method to rigorously derive conditions for the local stability of the Cournot–Nash equilibrium, and it complements the analytical findings with numerical simulations that highlight the emergence of periodic and chaotic dynamics once stability is lost. The additional exploration of heterogeneous cost parameters further enriches the study.

The topic is relevant, lying at the intersection of nonlinear dynamics, computational methods, and oligopoly theory. The results are both mathematically rigorous and economically insightful. I recommend acceptance, but some minor revisions are needed.

1. It appears that the CAD method introduced in Section 2 is not a new approach, but rather one that was developed in the last century by others (e.g., Collins and Hong). Although the authors make very good use of this method to prove their theorems (and I must admit that this proof technique is quite different from traditional approaches, which I had not seen before), I believe that the detailed description of the CAD method in Section 2 could be removed, with only appropriate references provided instead. This is because the paper is a research article rather than a textbook, and therefore it is not necessary to include a simple example to illustrate the computational procedure of the CAD method.

2. The language of the manuscript still requires further refinement in order to meet publication standards.

3. The formulas in the manuscript need to be reformatted, as their current presentation appears rather disorganized.

4. Perhaps the authors could clarify whether the symbolic computation method they employ is entirely error-free. Thus, the results of symbolic computation can be regarded as rigorous and sufficient for establishing mathematical theorems.

5. It is recommended that the authors discuss in the conclusion the potential applications of the CAD method in other areas of economics.

7. PLOS authors have the option to publish the peer review history of their article (what does this mean?). If published, this will include your full peer review and any attached files.

Reviewer #1: No

Reviewer #2: No

Reviewer #3: No

---

## [Editor Report · Decision Letter 2]

14 Oct 2025

Stability analysis of heterogeneous oligopoly games of increasing players: A computational approach

PONE-D-25-18321R2

Dear Dr. Li,

We’re pleased to inform you that your manuscript has been judged scientifically suitable for publication and will be formally accepted for publication once it meets all outstanding technical requirements.

Kind regards,

Pu-yan Nie

Academic Editor

PLOS ONE
---

## [Editor Report · Acceptance letter]

PONE-D-25-18321R2

PLOS ONE

Dear Dr. Li,

I'm pleased to inform you that your manuscript has been deemed suitable for publication in PLOS ONE. Congratulations! Your manuscript is now being handed over to our production team.

Kind regards,

on behalf of

Dr. Pu-yan Nie

Academic Editor

PLOS ONE